# Evidence for an internal model of friction when controlling kinetic energy at impact to slide an object along a surface toward a target

**Sylvain Famié**[1,2,3,4]*, **Mehdi Ammi**[4], **Vincent Bourdin**[3], **Michel-Ange Amorim**[1,2]

**1** Université Paris-Saclay, CIAMS, Orsay, France, **2** Université d'Orléans, CIAMS, Orléans, France,
**3** Université Paris-Saclay, CNRS, LIMSI, Orsay, France, **4** Université Paris 8, LIASD, Saint-Denis, France

\* sylvain.famie@universite-paris-saclay.fr

## Abstract

Although the role of an internal model of gravity for the predictive control of the upper limbs is quite well established, evidence is lacking regarding an internal model of friction. In this study, 33 male and female human participants performed a striking movement (with the index finger) to slide a plastic cube-like object to a given target distance. The surface material (aluminum or balsa wood) on which the object slides, the surface slope (-10˚, 0, or +10˚) and the target distance (25 cm or 50 cm) varied across conditions, with ten successive trials in each condition. Analysis of the object speed at impact and spatial error suggests that: 1) the participants chose to impart a similar speed to the object in the first trial regardless of the surface material to facilitate the estimation of the coefficient of friction; 2) the movement is parameterized across repetitions to reduce spatial error; 3) an internal model of friction can be generalized when the slope changes. Biomechanical analysis showed interindividual variability in the recruitment of the upper limb segments and in the adjustment of finger speed at impact in order to transmit the kinetic energy required to slide the object to the target distance. In short, we provide evidence that the brain builds an internal model of friction that makes it possible to parametrically control a striking movement in order to regulate the amount of kinetic energy required to impart the appropriate initial speed to the object.

## Introduction

When the bartender slides a shot glass across the bar to a customer, anticipating the effect of friction is crucial for controlling his/her movement. Although dealing with friction for sliding objects is not as common as when walking or driving on more or less slippery surfaces, unless one plays curling or hockey, friction is a force that the brain must consider for movement regulation in motor control. In this study, we provide some evidence that the brain builds an internal model of friction in order to control parametrically the upper limb's kinetic energy when sliding an object on a surface toward a target distance.

The role of an internal model of gravity in motor control is quite well established in neuroscience [1], using pointing [2,3] or interceptive tasks [4,5], allowing predictive control of movement by anticipating the effect of gravity on the body and/or falling objects. Some

**Funding:** The author(s) received no specific funding for this work.

**Competing interests:** The authors have declared that no competing interests exist.

neuroimaging evidence points to the role of the vestibular cortex when activating this internal model in perceptual judgments [6]. Similarly, the cerebellum appears to build an internal model of the sensory consequences of gravity during passive self-motion, and of load force acting on the skin for the predictive control of grip force when lifting objects [7–9]. In the latter case, it has been shown that friction between the fingers and objects contributes to the feedforward control of both the grip force and upper limb acceleration when adjusting load force during object transport in order to prevent slipping [10]. Moreover, it has been shown that vision influences the adjustment of grip-load force coupling by estimating force based on visual motion signals [11], and that the brain relies on visual input more than tactile input to estimate friction to prevent falls when standing on inclined surfaces [12]. Finally, the literature on dexterous manipulation suggests that the appropriate control of grip force may be informed initially by adapting internal models built from previous experience in various contexts, and adjusted flexibly on the basis of sensory feedback [13,14].

Additional literature suggesting an internal model of friction comes from perceptual studies showing that the last memorized position of a moving object is influenced by the implied gravity or friction in the visual scene [15,16]. Likewise, Amorim et al. [17] used a perception of causality paradigm [18] in which an object A (launcher) moves toward a stationary object B (target), then when the launcher reaches the target, the latter is set into motion and the launcher becomes stationary. Participants indicated where the target object (sliding on more or less inclined surfaces) should have stopped after colliding with the launcher. The authors applied classical mechanics equations on the responses to compute the subjective value of the friction coefficient for the target, assuming that friction would cause it to decelerate post-collision. Their results suggest that our internal model of the coefficient of friction is consistent for horizontal and upward slopes, but overestimated for downward slopes. One might wonder if these results could be generalized to the parameterization of a ballistic gesture intended to slide an object along a given surface to a target distance.

Literature on the predictive control of hand grip shows that the brain predicts the sensory consequences of various forces (friction, reaction force, load, etc.) not only to prevent the objects we hold from falling [19–21] but also to regulate the amount of kinetic energy in tasks such as golf putting [22] or stone knapping [23]. Assuming that striking movements correspond to a specific generalized motor program [24,25] inherited from evolution [26,27], we used several experimental manipulations (surface slope, surface material and target distance) in this study to investigate the role of the internal model of friction in the parameterization of a ballistic gesture to slide an object to a target distance. Here, we hypothesized that for each surface material, the participants would initially calibrate this internal model on the basis of online feedback error learning [28,29] involving well-established brain networks for movement adaptation, such as the cerebellum and the parietal cortex. Then, they would internalize this "pretty good" (i.e., functionally relevant, although not perfect) internal model [30] of the coefficient of friction across task constraints (surface slope, target distance) with the same surface material. Finally, we tested the hypothesis that the brain regulates parametrically the kinetic energy of the upper limb segments as well as of the launched object, based on visual control variables such as object initial speed and spatial error. Moreover, we tested the assumption that motor redundancies [31,32] would allow for variability in motor coordination to reach the required kinetic energy at impact.

## Materials and methods

Thirty-three people (22 men and 11 women) with a mean age of 25 years (SD = 4.99) participated in this experiment. All of the subjects were right handed (Edinburgh test, [33]), had

normal or corrected-to-normal vision, and reported no physical injury or pathology that could affect hand movement. The experiment was approved by the local "Comité d'éthique de la recherche de l'Université Paris-Saclay" ethics committee (CER-Paris-Saclay-2018-021-R). After reading the instructions, the participants signed a consent form and filled in the Edinburgh questionnaire.

## Experimental setup

During the experiment, the participants were asked to strike an object with their index finger to cause it to slide to a target distance. The object was a rectangular parallelepiped (length = 60 mm, width and height = 50 mm), printed in PLA (polylactide), weighing 46 g. A circle target was drawn on the center of the object face to be struck by the index finger (see Fig 1). For the sake of simplicity, we will call this object the "cube" from now on.

At the start of each trial, the cube was positioned at one entrance of a gutter setup with a square section allowing the cube to slide along a single axis of translation while preventing rotations (Fig 1). The gutter's inner dimensions were: length = 800 mm, width = 55 mm and height = 25 mm. The lateral sides of the gutter were made of aluminum. However, the bottom sliding surface material was either balsa wood or aluminum, depending on the condition.

The participants were instructed to strike the cube in order to cause it to reach one of two target distances: 25 cm or 50 cm, indicated by a sticker positioned on the upper right side of the gutter. The gutter setup was fixed on a motorized table allowing the experimenter to vary surface inclinations (-10˚, 0˚, +10˚) and to adjust the setup height to ensure the index finger start position was on the cube's target face (see Fig 1).

## Data acquisition

During the experiment, a nine-camera OptiTrack motion capture system (Model S250e) recorded the arm and cube movements at 250 Hz. Reflective markers were placed on the setup, cube and right upper limb. As illustrated in Fig 1, 4 mm markers were positioned on the head of the phalanx of the index finger and on the metacarpals on the second and third fingers

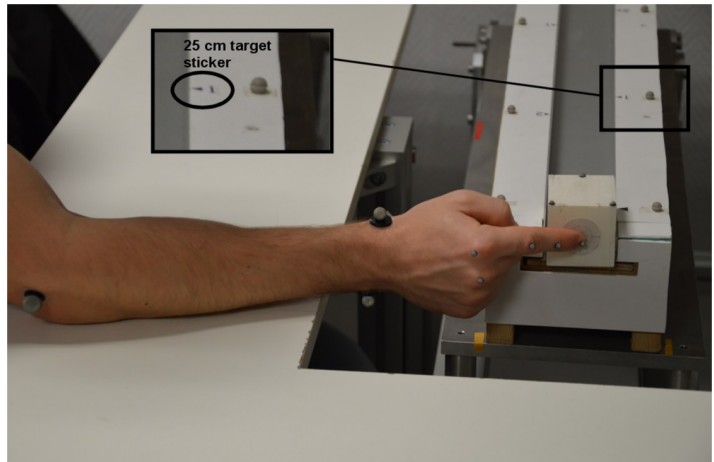

**Fig 1. Experimental setup and task.** Participants were instructed to strike a cube to slide it to one of two target distances indicated by a triangular sticker (together with a reflective marker) positioned on the upper right side of a gutter). Here, we illustrate the initial standard position of the hand with respect to the experimental setup in the 0˚ slope condition with an aluminum surface material. Reflective markers were placed on the setup, cube and right upper limb of the participant, for the purpose of motion analysis.

in accordance with Zhang et al. [34]. Markers of different sizes were placed on the back of the hand, wrist and elbow following International Society for Biomechanics (ISB) recommendations for reporting human joint motion [35]. For the wrist, 11 mm reflective markers were placed on the most caudal points on the styloid process of the radius and the ulna. For the elbow, 14 mm reflective markers were placed on the most caudal points on the lateral and medial epicondyle. Finally, a cluster of three 4 mm reflective markers was placed on the cube, and 7 mm reflective markers were arranged along the edges of the setup to indicate the initial and target (25 cm and 50 cm) distances.

## Experimental task

Before the start of the experiment, the participants stretched out their arm for measurement of forearm length ($L_{Forearm}$) and hand length ($L_{Hand}$). The height and weight of each participant were also collected. Then, before each trial, the participants placed their elbow at a standard position (determined with the participant to ensure it was comfortable) indicated by a sticker on the table to allow them to pivot their forearm while keeping their elbow on the table. The participants were also told to keep their forearm more or less perpendicular with respect to the setup, with the distal phalanx of their index finger in front of the cube target face (see Fig 1). Their seating position was adjusted to allow their elbow to rest on the table with the arm and the forearm roughly perpendicular to each other. During the striking movement, the participants were free to adopt any motor coordination they wished as long as the elbow remained in the same position on the table. The required task was to strike the cube, causing it to slide along the gutter until the front edge of the cube stopped at the target distance indicated by a triangular sticker and a reflective marker positioned on the gutter (see Fig 1). After each shot, the experimenter returned the cube to its standard initial position, with 5 cm of the cube resting on the surface and 1 cm protruding from the gutter to avoid collision between the hand and the setup.

There were 12 conditions resulting from the combination of two target distances (25 cm and 50 cm), two surface materials (balsa wood and aluminum), and three surface slopes (-10˚, 0˚ and +10˚). In total, the participants performed 120 trials with ten repetitions per condition. The surface slope trials were organized in blocks, with each experimental session starting with the 0˚ condition, followed by either the -10˚ slope then the +10˚ slope ("0˚, -10, +10" block order group), or by the +10˚ slope then the -10˚ slope ("0˚, +10, -10" block order group). These three blocks of trials were with either the balsa wood surface, followed by three blocks of trials on the aluminum surface, or vice versa, with the order being counterbalanced across participants. Finally, all of these conditions (60 trials) were run in one block of trials for the 25 cm target distance, followed by a block for the 50 cm target distance (60 trials). These different block orders resulted in four groups as illustrated in Table 1.

## Data analysis

Motion capture was recorded using AMASS software, and data was processed with MATLAB homemade routines to generate the files necessary for running statistics using STATISTICA and SPSS. No low-pass filter was applied to the 3D marker position signals because, if the filter parameters commonly used in the human movement literature were applied (e.g., 10–15 Hz cutoff, second or third order Butterworth), the collision phenomenon (lasting about 15 ms) would have been considered noise and filtered out.

We segmented the arm movement in three phases on the basis of both the index fingertip and cube kinematics as illustrated in Fig 2, with speed measured along the y-axis (aligned with the gutter main axis). The speed was positive for motion towards the cube, and negative for

**Table 1. Description of the block order used in the Experiment in each group, whether A) the "0˚, -10˚, +10˚" block order group, or B) the "0˚, +10˚, -10˚" block order group.** Each block order group was subdivided in two subgroups: One beginning with aluminum (alu) followed by balsa wood (balsa), and the other in the opposite order.

| A | 0˚, -10˚, +10˚ block order group | | | | | | | | | | | |
|---|---|---|---|---|---|---|---|---|---|---|---|---|
| Target distance | 25 cm | | | | | | 50 cm | | | | | |
| Slope | 0˚ | -10˚ | +10˚ | 0˚ | -10˚ | +10˚ | 0˚ | -10˚ | +10˚ | 0˚ | -10˚ | +10˚ |
| Block number | 1 | 2 | 3 | 4 | 5 | 6 | 7 | 8 | 9 | 10 | 11 | 12 |
| Group 1 | Alu | Alu | Alu | Balsa | Balsa | Balsa | Alu | Alu | Alu | Balsa | Balsa | Balsa |
| Group 2 | Balsa | Balsa | Balsa | Alu | Alu | Alu | Balsa | Balsa | Balsa | Alu | Alu | Alu |
| B | 0˚, +10˚, -10˚ block order group | | | | | | | | | | | |
| Target distance | 25 cm | | | | | | 50 cm | | | | | |
| Slope | 0˚ | +10˚ | -10˚ | 0˚ | +10˚ | -10˚ | 0˚ | +10˚ | -10˚ | 0˚ | +10˚ | -10˚ |
| Block number | 1 | 2 | 3 | 4 | 5 | 6 | 7 | 8 | 9 | 10 | 11 | 12 |
| Group 3 | Alu | Alu | Alu | Balsa | Balsa | Balsa | Alu | Alu | Alu | Balsa | Balsa | Balsa |
| Group 4 | Balsa | Balsa | Balsa | Alu | Alu | Alu | Balsa | Balsa | Balsa | Alu | Alu | Alu |

motion away from the cube. The initial arming phase started when the participants moved their index fingertip back away from the cube, and ended when the arm stopped at its maximal extension with speed equal to zero. During the striking movement, the hand moved towards the cube with increasing speed until impact. The start of the cube sliding phase (see Fig 2) corresponded to the index-cube initial contact time point ($t_{contact}$). It was determined using the Multiple Sources of Information method (MSI-method, see [36,37]) using the fingertip deceleration peak (due to the collision) and time interval between max fingertip speed ($t_{maxFS}$) and minimum cube speed ($t_{minCS}$) just before the cube started moving. The cube's movement onset was defined as cube speed greater than 0.04 m/s. Accordingly, the contact time point ($t_{contact}$) was determined for the [$t_{maxFS}$; $t_{minCS}$] interval as:

$$t_{contact} = min\left\{ t\left(\frac{Acceleration}{Min\ Acceleration}\right) > 0.4 \right\} - dt \tag{1}$$

Each fingertip acceleration value during the time interval (delimited by the curly brackets in the formula) was normalized by dividing each value by the minimum acceleration value observed within the interval. When fingertip deceleration was greater than 40% (cf. the 0.4 value in the formula), the immediately preceding time point (i.e., d$t$ in the formula) was considered to be the initial time of contact. We used this criterion in order to avoid false detection of finger-cube contact, because after maximum finger speed (beginning of the time interval of interest) there was sometimes a slight deceleration (less than 40%) in finger movement before contact.

At the start of the cube sliding phase, the initial brief finger-object contact duration concomitant with a mutual speed change reflected energy transfer from the finger to the cube which started to slide. This brief duration is delimited by the dashed and dotted vertical lines in the cube sliding phase panel of Fig 2. From there, cube kinematics were used to determine the coefficient of kinetic friction ($\mu_K$) and optimal cube speed in order to reach the target distance in each condition. μk was computed from:

$$\mu_K = \frac{\left(\frac{dec}{-g}\right) - \sin(\alpha)}{\cos(\alpha)} \tag{2}$$

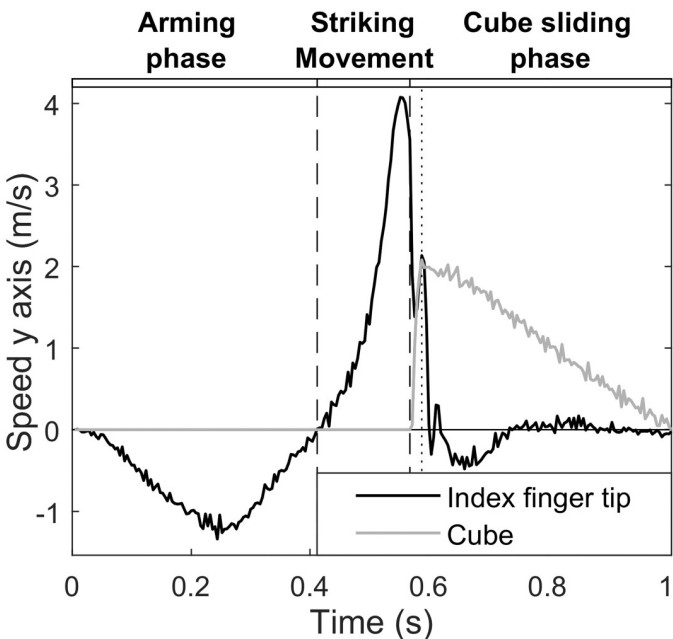

**Fig 2. Segmentation of the ballistic gesture and cube movement.** Three phases were taken into consideration on the basis of both index fingertip and cube speed along the main axis of the experimental setup (the gutter). The initial arming phase started when the participants moved their index fingertip back away from the cube, and ended when the arm stopped at its maximal extension with speed equal to zero. During the striking movement, the hand moved towards the cube with increasing speed until impact. The start of the cube sliding phase corresponded to the index-cube initial contact duration concomitant with a mutual speed change reflecting energy transfer from the finger to the cube which started to slide.

To estimate μk, we needed to estimate cube deceleration as a function of gravity ($g$) and surface slope ($\alpha$). This was done by fitting a second-order polynomial function to the cube position across time (i.e., from the cube's initial maximum speed until it stopped), with $y = Ax^2 + Bx + C$. Then, cube deceleration *dec* was computed from the second derivative, $dec = y'' = 2A$. This μk value was determined for each experimental trial. In this study, $\mu_K$ takes into account both of the forces that decelerate the cube: surface material and the aluminum sides of the gutter. In order to validate the fact that the two surface materials (aluminum and balsa wood) have different coefficients of kinetic friction that would modulate behavior, we computed mean $\mu_K$ for each surface material for each participant. A paired *t* test revealed a significantly greater $\mu_K$ for the balsa wood surface ($M = 0.47$, SE = 0.01) compared to the aluminum surface ($M = 0.37$, SE = 0.01), $t(32) = 16.99$, $p < 0.05$.

Then, we estimated the optimal cube speed (Cube Speed_optimal) to send the object to the target distance ***d*** for each experimental condition, using:

$$\text{Cube Speed}_{\text{optimal}} = \sqrt{2\,g\,d\,[\sin(\alpha) + \mu_K\,\cos(\alpha)]} \tag{3}$$

These Cube Speed_optimal values are summarized in Table 2 for each experimental condition under increasing task constraints.

## Performance variables

Spatial error and initial cube speed (m/s) were used as performance variables. Spatial error was defined as the difference in distance between the target distance and the final position of the

**Table 2. Mean optimal cube speed (m/s) for each experimental condition with ascending task constraint.**

| Target distance | 25 cm | 25 cm | 25 cm | 50 cm | 25 cm | 25 cm | 50 cm | 25 cm | 50 cm | 50 cm | 50 cm | 50 cm |
|---|---|---|---|---|---|---|---|---|---|---|---|---|
| Surface slope | -10˚ | -10˚ | 0˚ | -10˚ | 0˚ | +10˚ | -10˚ | +10˚ | 0˚ | 0˚ | +10˚ | +10˚ |
| Surface material | Alu | Balsa | Alu | Alu | Balsa | Alu | Balsa | Balsa | Alu | Balsa | Alu | Balsa |
| Optimal cube speed | 0.967 | 1.191 | 1.347 | 1.368 | 1.518 | 1.625 | 1.684 | 1.767 | 1.905 | 2.147 | 2.297 | 2.499 |

cube's front edge as a percentage of target distance, with positive values for overshoots and negative values for undershoots. Motor adaptation was investigated by fitting an exponential curve to the trial factor data for both performance variables with the following equation:

$$\text{Performance} = \textit{offset} + \textit{constant} * \exp(-(\text{trial} - 1)/\textit{tau}) \qquad (4)$$

*Offset* quantifies final performance after the adaptation process. *Constant* quantifies the magnitude and direction of performance variation. *Tau* quantifies the half-life of exponential decay, in terms of trial number. Only exponential fits with a $R^2 > 0.75$ were considered as reflecting meaningful adaptation and were displayed in the figure for mean performance and motor adaptation.

In order to test whether there were systematic spatial errors in the first trial of each condition that would be reduced across trials, we first performed an ANOVA on spatial error in trial 1 for significant interactions involving the Trial and Surface material factors. Then, we compared mean spatial error to 0%, for the first and last trials in order to test if motor adaptation led to an optimal performance. Accordingly, we used Bonferroni corrected $t$ tests with $\alpha$ = 0.05/6 = 0.008 (see Fig 3) or $\alpha$ = 0.05/4 = 0.0125 (see Fig 4), depending on the number of initial and final trials for a given surface material, illustrated by an asterisk when significant.

As mentioned in the *Experimental task* section, the order of the surface slope blocks differed depending on the group. For one group, the 0˚ condition was followed by the -10˚ condition ("0˚, -10˚, +10˚" block order group) whereas it was followed by the +10˚ condition for the other group ("0˚, +10˚, -10˚" block order group). Therefore, we conducted additional analyses in order to test more deeply the generalization of subjective μ from one block to another for a given surface material. Evidence in favor of this generalization of subjective μ is tagged with thumb-up symbols in the figures of the *Performance and motor adaptation as a function of block transition* section. This was the case when the spatial error for the last trial of a block and the first trial of the following block did not differ from zero. Accordingly, we expected no sign of motor adaptation (as measured by the exponential fits) in neither spatial error nor the initial cube speed for the new block. Note that by displaying our data in separate plots depending on group and distance, the data points across trials became noisier. As a consequence, in order to increase our sensitivity in detecting motor adaptation, we decreased the $R^2$ threshold to 0.50 (instead of 0.75). Similarly, to avoid false alarms when stating that spatial error was significantly different from zero for initial and final trials, we used Bonferroni corrected $t$ tests with $\alpha$ = 0.05/12 = 0.004 as the significance criterion (12 being the number of comparisons per surface material). Finally, in order to assert statistical evidence in favor of the generalization of subjective μ from one block to the next, we compared performance between the last trial of a block and the initial trial of the next block using Bonferroni corrected $t$ tests (with $\alpha$ = 0.05/4 = 0.0125 as a criterion, the four comparisons corresponding to the transition between the first and second block, and between the second and third block, for each level of Target distance). Such significant changes in performance between blocks are illustrated by an arrow in Figs 5 and 6.

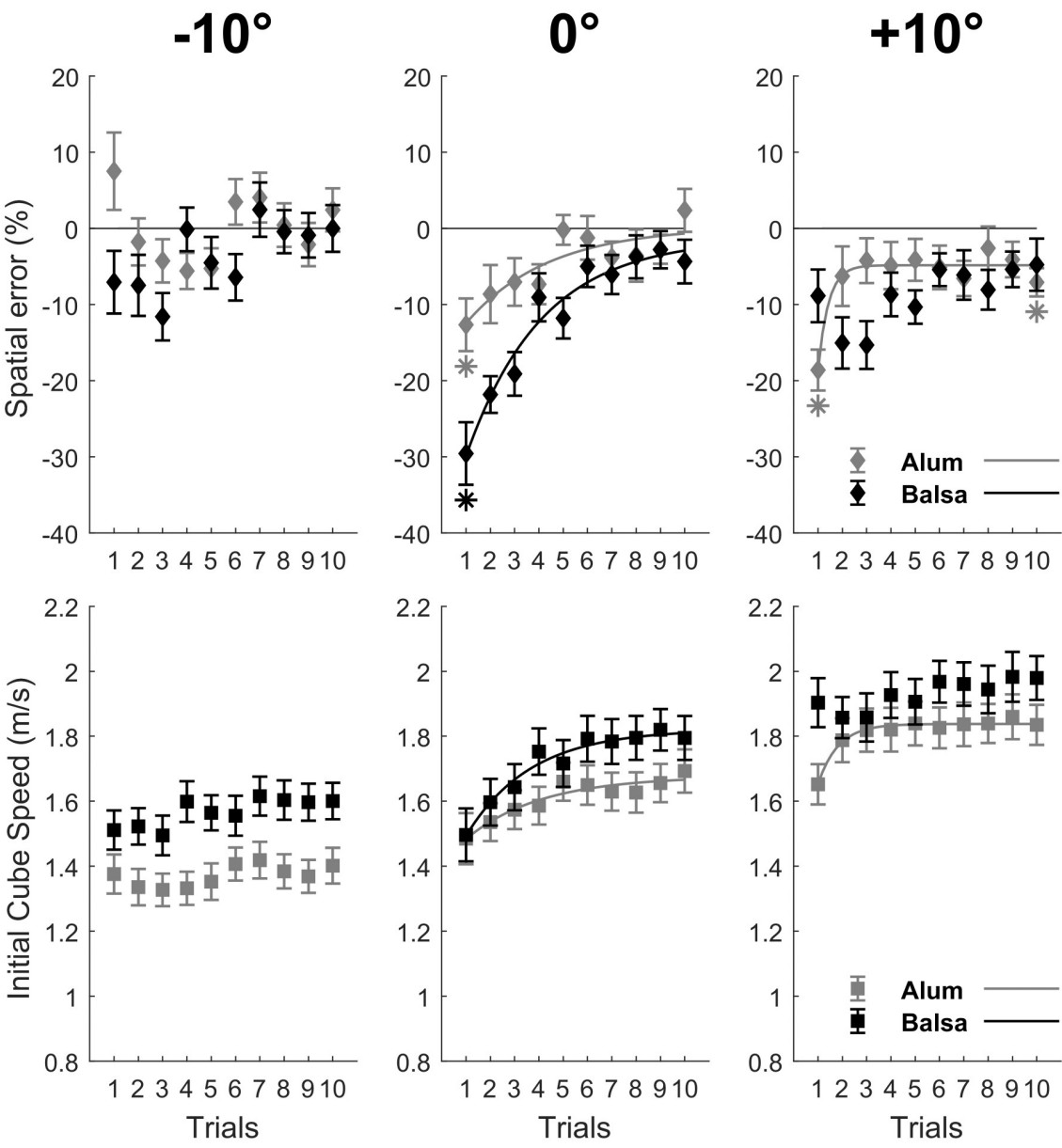

**Fig 3. Performance and motor adaptation as a function of surface material and inclination.** The upper panels show motor adaptation only in the 0° and +10° condition in the spatial error data (as a percentage of target distance), as a consequence of variation in initial cube speed illustrated in the lower panels. Exponential fits with $R^2 > 0.75$ illustrate motor adaptation across trials. Asterisks for spatial error (mean ± SE) indicate initial and final means differing significantly (Bonferroni correction with $\alpha = 0.05/6 = 0.008$ for each surface material) from zero.

The rationale is that if participants reached optimal performance (spatial error not significantly different from 0) at the end of a block, this would reflect the fact that they had converged to the correct value of friction within a block. Accordingly, we theorized that participants should be able to parameterize their movement despite a change in block condition, such as a change of surface slope and/or target distance, in order to reach optimal performance right from the first trial of the new block condition. If not, we expected signs of motor adaptation in the new block of trials suggesting that a recalibration of subjective μ across trials was needed.

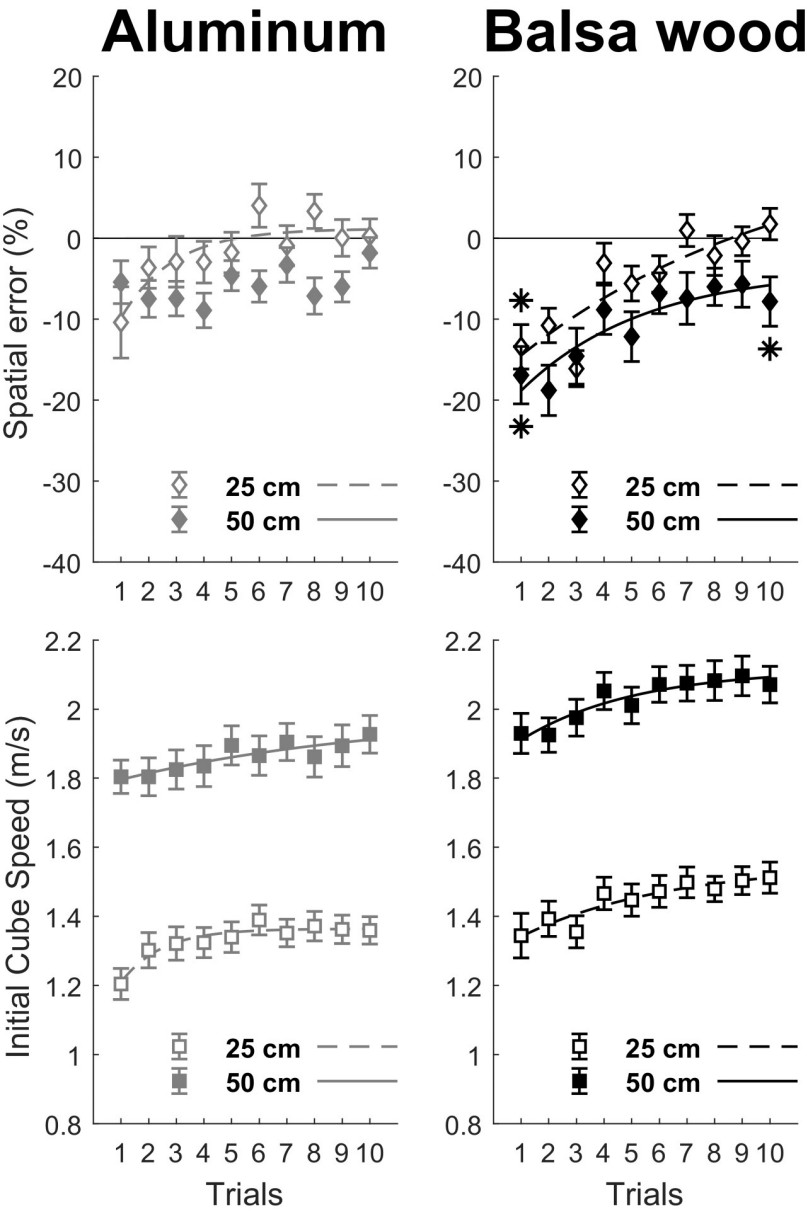

**Fig 4. Performance and motor adaptation as a function of surface material and target distance.** Motor adaptation across trials is observed mainly for the balsa wood surface material (right panels) corresponding to more demanding conditions in order to counteract the effect of friction on cube displacement. Exponential fits with $R^2 > 0.75$ are illustrated, together with asterisks when initial and final mean spatial error differed significantly (Bonferroni correction with $\alpha = 0.05/4 = 0.008$ for each surface material) from zero.

However, traces of a generalization of the internal model of physics (including subjective μ) from one block to the next could also manifest in a greater learning speed in the new block.

Motor coordination was first examined geometrically on the basis of joint angular amplitude between the beginning and end of the striking movement as projected onto the xy (horizontal) plane, corresponding to the table on which the elbow rested. The angular amplitudes of interest were: forearm angle $\theta_{Forearm}$, wrist angle $\theta_{Wrist}$ and index finger angle $\theta_{Index}$, at the last (10th trial of each condition). The forearm angle $\theta_{Forearm}$ is formed by the y axis (aligned

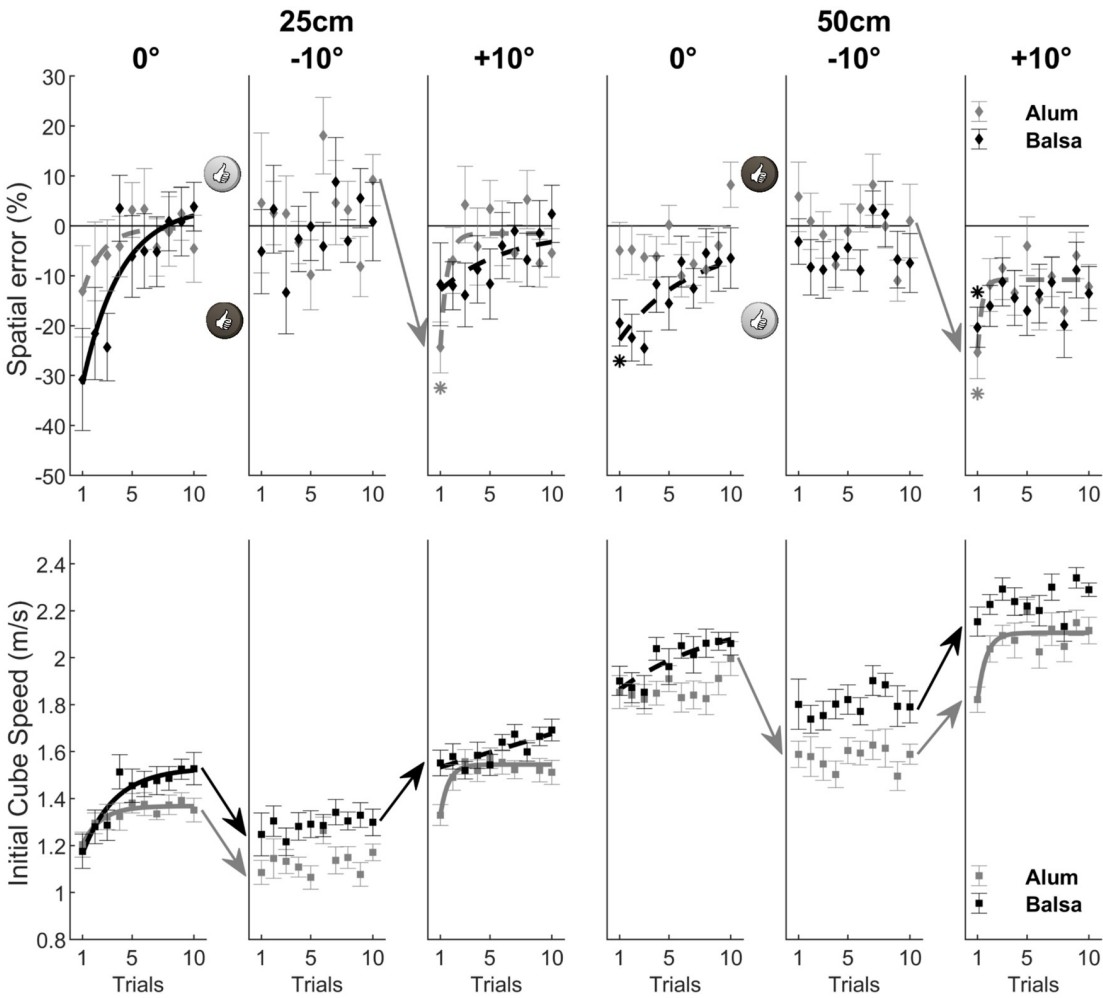

**Fig 5. Performance and motor adaptation as a function of surface material, surface slope and target distance, is illustrated for the "0˚, -10˚, +10˚" block order groups.** Evidence in favor of generalization of subjective μ between blocks with a given surface material is tagged with thumb-up symbols in the upper panel. This generalization occurred when the spatial error for both the last trial of a block and the first trial of the following block did not differ from zero (Bonferroni correction with α = 0.05/12); asterisks illustrate evidence against the generalization hypothesis. Exponential fits with $R^2 > 0.75$ (continuous lines) and $R^2 > 0.50$ (dashed lines) indicate strong and moderate evidence of motor adaptation, respectively. Arrows illustrate significant (Bonferroni correction with α = 0.05/4) change in behavior between blocks. The results show motor adaptation at the initial block of trials corresponding to a 0˚ slope and a 25cm target distance, whether on an aluminum or balsa wood surface depending on group (see Table 1). This calibration of subjective μ for a given surface material in the 0˚ condition led to an appropriate re-parameterization of the movement in the following -10˚ condition (see the thumb-up symbols for the 0˚ to -10˚ transitions), also illustrated in an adequate change in initial cube speed. In contrast, motor adaptation was needed for the -10˚ to +10˚ block transition.

with the gutter) and the forearm axis (joining the elbow and the wrist central joint). The wrist central joint was estimated as the projection of the radius styloid marker onto the xy plane. The wrist angle $\theta_{Wrist}$ is formed by the intersection between the forearm axis and the hand axis (joining the wrist central joint and the third metacarpophalangeal head). The index finger angle $\theta_{Index}$ is formed by the hand axis and the index finger axis (joining the second metacarpophalangeal head and the fingertip marker).

Last, we computed the kinetic energy (KE) at impact for the upper limb and the cube. We estimated total upper limb KE (Total KE) as the sum of Forearm KE and Hand KE using the

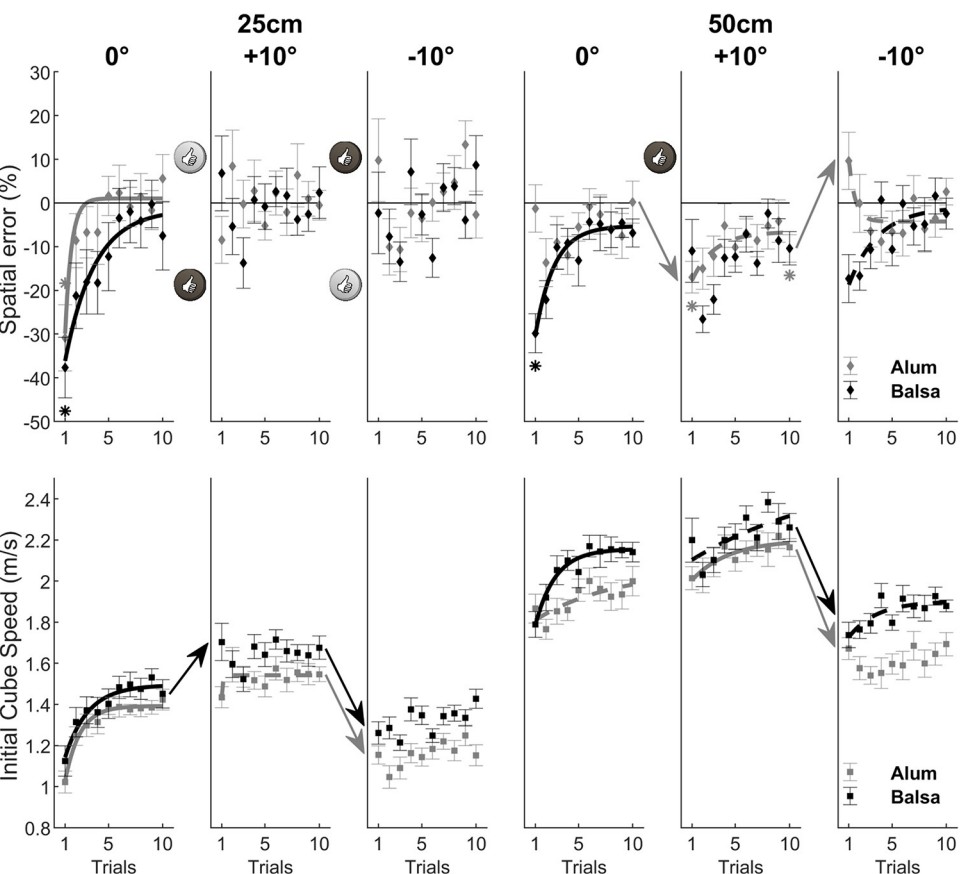

**Fig 6. Same as Fig 5, for the "0˚, +10˚, -10˚" block order groups.** Again, the results show motor adaptation at the initial block of trials leading to a subjective μ calibration that allowed for adequate movement re-parameterization in the following +10˚ conditions (see 0˚ to +10˚ transition), except for 50 cm with aluminum. Then, movement re-parameterization for the -10˚ to +10˚ transition was successful (see the thumb-up symbols) only in the shorter target distance condition (25 cm).

following equations adapted from Winter [38]:

$$KE_{Forearm} = {}^{1}\!/_{2}\, m_{Forearm} \left\{ \left[ - L_{Forearm} \sin(\theta_{Forearm})^2 + L_{Forearm} \cos(\theta_{Forearm}) \right]^2 \right\} + {}^{1}\!/_{2}\, I_{Forearm}\, \theta_{Forearm}{}^2 \quad (5)$$

$$KE_{Hand} = {}^{1}\!/_{2}\, m_{Forearm} \left\{ \left[ -L_{Elbow} \sin(\theta_{Elbow}) - L_{Wrist} \sin(\theta_{Wrist})^2 + L_{Elbow} \cos(\theta_{Elbow}) + L_{Wrist} \cos(\theta_{Wrist}) \right]^2 \right\}$$
$$+ {}^{1}\!/_{2}\, I_{Wrist}\, \theta_{Wrist}{}^2 \quad (6)$$

$$Total\ KE = KE_{Forearm} + KE_{Hand} \quad (7)$$

In Eqs 5 and 6, the segment lengths (L) were reported following the ISB recommendation [35], whereas the anthropometric parameters (limb segment mass m and inertia I) were derived from de Leva [39] anthropometric tables. Last, we computed cube KE after impact as:

$$KE_{Cube} = {}^{1}\!/_{2}\, m_{Cube}\, Speed_{Cube}{}^2 \quad (8)$$

Since the cube is blocked laterally in a gutter, only the translational part of KE was taken into account.

## Results

### Mean performance and motor adaptation

In this study, we assume that participants have an accurate internal model of gravity (as a physical invariant) and an accurate perception of surface slope. Therefore, if participants have a "pretty good" internal model of physics that links the initial cube speed to the final spatial error, spatial error should directly reflect participants' internal model of μ (subjective μ). A subjective μ consistent with the physical μ would lead to no spatial error if participants are able to regulate their movement to impart the appropriate cube speed. Moreover, we investigated motor adaptation by examining how the effect of trial on performance varied with the other experimental factors. As a consequence, although these factors showed main effects on performance, $p$s < 0.001, we focused our analysis on interactions involving the trial factor. Repeated measures ANOVAs on the performance variables were performed with the following within-subjects factors: two target distances (25 cm and 50 cm) x two surface materials (balsa wood and aluminum) x three surface slopes (-10˚, 0˚ and +10˚) x ten trials. Although ANOVA is robust with respect to violations of normality in terms of Type I error [40], Q-Q plots showed that our performance variables data appear to be normally distributed.

The fourth-factor interaction between Surface material, Slope, Target distance, and Trial was not significant, for neither spatial error, $F(18, 576) = 0.44$, $p = 0.98$, nor initial cube speed, $F(18, 576) = 0.48$, $p = 0.97$. However, we found a significant three-factor interaction between Surface material, Slope and Trial on spatial error, $F(18, 576) = 2.36$, $p = 0.0013$, and on initial cube speed, $F(18, 576) = 2.19$, $p = 0.003$, as illustrated in Fig 3. We first wanted to test whether there were systematic spatial errors in the first trial of each condition. For this, we performed an ANOVA on spatial error in trial 1 with Surface material and Slope as within-subject factors. It showed that spatial error varied significantly as a function of Surface material ($M_{alu}$ = -7.92%, $M_{balsa}$ = -15.16%), $F(1, 32) = 6.14$, $p = 0.019$, and of Slope ($M_{-10˚}$ = 0.23%, $M_{0˚}$ = -21.11%, $M_{+10˚}$ = -13.73%), $F(2, 64) = 15.23$, $p < 0.001$. Finally, there was a significant interaction between Surface material and Slope, $F(2, 64) = 12.60$, $p < 0.001$. Bonferroni corrected $t$ tests (with α = 0.05/6) showed that this initial spatial error differed significantly from 0% for both surface materials in the 0˚ condition (Alu: $t(32) = 3.66$, $p = 0.0009$; Balsa: $t(32) = 7.18$, $p < 0.001$) and for aluminum in the +10˚ condition ($t(32) = 6.92$, $p < 0.001$), as highlighted by the asterisks in Fig 3. Interestingly, spatial error was significantly greater for Balsa wood ($M$ = -29.57%) compared to Aluminum ($M$ = -12.66%), $t(32) = 3.40$, $p = 0.002$, in the 0˚ slope condition. This initial difference was due to a similar initial cube speed, $t(32) = 0.26$, $p = 0.80$, that induced a different error due to the different coefficients of friction. Given that participants always started a surface material and target distance blocks with the 0˚ condition, keeping cube speed constant may have facilitated the estimation of μ (from the visual consequences of the cube kinematics), and the subsequent parameterization of the movement in order to reduce spatial error across subsequent trials.

The parameterization of the movement is illustrated in the significant exponential curves ($R^2$ > 0.75, see Methods), for spatial error or initial cube speed illustrated in Fig 3 for the aforementioned three sub-conditions where spatial error in trial 1 differed significantly from 0%. On the other hand, the lack of an exponential fit may reflect an exploratory strategy, during which subjects try out different ways of hitting the object until they find the one that works 'best', as in the -10 condition for both surface materials, and in the +10˚ condition for balsa. Finally, we wanted to test whether performance was optimal in the last trial of each condition of the interaction between Surface material and Slope on spatial error. Bonferroni corrected $t$ tests (with α = 0.05/6) showed that spatial error in trial 10 differed significantly from 0% only for aluminum in the +10˚ slope condition ($M$ = -7.08%), $t(32) = 3.82$, $p = 0.0006$. As we will

show in subsequent analyses taking into account block order groups, this suboptimal final performance was due to the 50 cm target distance in the "0˚, +10˚, -10" block order group.

The repeated measures ANOVA also showed a significant interaction between Surface material, Target distance and Trial, $F(9, 288) = 1.93$, $p = 0.048$, for spatial error (see Fig 4). An initial ANOVA on spatial error in trial 1, with Surface material and Target distance as within-subject factors, showed that, although spatial error did not vary with Target distance ($M_{25cm}$ = -11.91%, $M_{50cm}$ = -11.17%), $F(1, 32) = 0.07$, $p = 0.789$, it did vary as a function of Surface material ($M_{alu}$ = -7.92%, $M_{balsa}$ = -15.16%), $F(1, 32) = 6.14$, $p = 0.019$. Moreover, both factors did not interact on spatial error, $F(1, 32) = 1.68$, $p = 0.20$. Bonferroni corrected $t$ tests (with $\alpha$ = 0.05/4) showed that spatial error in trial 1 differed significantly from 0% only for balsa wood (25 cm condition: $t(32) = 3.78$, $p < 0.001$; 50 cm condition: $t(32) = 6.17$, $p < 0.001$), as highlighted by the asterisks in Fig 4.

The signature of adaptation in the exponential modulation of motor behavior (with $R^2 > 0.75$), whether spatial error or initial cube speed, was observed in each condition except for spatial error in the 25 cm Target distance condition for aluminum (see Fig 4). However, in this sub-condition the lack of an exponential fit might reflect an exploratory strategy to find the best way of hitting the object. Finally, Bonferroni corrected $t$ tests (with $\alpha$ = 0.05/4) showed that spatial error in trial 10 differed significantly from 0% only for balsa wood in the 50 cm target distance condition, $t(32) = 4.03$, $p < 0.001$. Although we could not evidence a fourth-factor interaction between Surface material, Slope, Target distance, and Trial, for neither spatial error nor initial cube speed, the next analyses will examine the effect of these factors in relation to block order group in order to investigate the generalization of subjective μ from one block to another.

## Performance and motor adaptation as a function of block transition

As explained in the *Performance variables* section of *Materials and Methods*, we conducted additional analyses as a function of block order groups in order to test the generalization of subjective μ from one block to another for a given surface material. The spatial error panels of Figs 5 and 6 show numerous instances of generalization suggesting that participants were able to re-parameterize their movement from one block to another on the basis of the internal model of μ for a given surface. On the other hand, in other cases (no thumb up in block transition), they required additional motor adaptation to better tune the subjective μ value with respect to the physical invariants of the task and thereby improve movement parameterization. A brief summary of these results is provided in the captions of Figs 5 and 6. Overall, the results for both groups (see Figs 5 and 6) show that it is easier to generalize the subjective μ value tuned in the 0˚ condition when the next block is a -10˚ or a +10˚ slope. In contrast, the -10˚ to +10˚ block transition ("0˚, -10˚, +10˚" block order group) requires motor adaptation and a recalibration of subjective μ, although participants increased initial cube speed (but not enough) in order to cope with the increased task constraint. For example, optimal cube speed values (see Table 2) show that, when changing from -10˚ to +10˚ for the 50 cm target distance, one needs to increase cube speed by about 50% in the balsa wood condition (i.e., from 1.68 m/s to 2.50 m/s) and by about 70% in the aluminum condition (i.e., from 1.37 m/s to 2.30 m/s). Still, as previously mentioned (see *Performance variables* section), greater learning speed (for motor adaptation) in the +10˚ block compared to the 0˚ block for aluminum (see Fig 5) suggests partial generalization of the internal model of μ between blocks of the same surface material. Finally, the +10˚ to -10˚ block transition ("0˚, +10˚, -10˚" block order group) shows good re-parameterization of the movement for the 25 cm target distance based on the subjective μ tuned in the initial block, whereas additional tuning (through motor adaptation) was needed in the 50 cm condition (see Fig 6).

Some blocks were especially challenging in terms of movement re-parameterization. Indeed, as described in Table 1, once participants performed the six blocks in the 25 cm target distance condition, the surface material was changed in the next six blocks in the 50 cm target distance condition. So, the transition between block 6 and block 7 involved a change in three physical properties of the stimulus, i.e., surface slope, surface material and target distance. Fig 7 illustrates the change in performance between the last trial of block 6 (last trial of the 25 cm blocks) and the first trial of block 7 (first trial of the 50 cm blocks). We used Bonferroni corrected ($\alpha = 0.05/4$) $t$ tests to examine if there was a significant performance change between those two trials (illustrated by arrows in Fig 7), and a significant spatial error with respect to zero (illustrated by asterisks in Fig 7). We theorized that performance not differing significantly from zero after a change in three physical properties of the stimulus would reflect adequate movement re-parameterization (in order to impart optimal initial cube speed). The results show appropriate movement re-parameterization despite the dramatic change in the stimuli, except for the more demanding transition (see the triangle data points in Fig 7) requiring an increase of about 120% of initial cube speed (see Table 2 optimal initial cube speed: 0.967 m/s for a 25 cm target distance on a -10˚ aluminum surface, to 2.147 m/s for a 50 cm target distance on a 0˚ balsa wood surface).

## Motor coordination parameterization

The participants performed the task with their elbow resting on the table. Therefore, the movement involved primarily rotations of the forearm, wrist and index finger. The increase in task constraints (due to the surface slope, target distance and surface material) corresponded to a continuum of optimal initial cube speed (see Table 2). Therefore, we assumed that the striking movement parameterization would reflect this continuum. However, in order to perform the task under increasing constraints, the main limb segment masses contributing to the task were clearly the forearm and the hand (see section *Motor coordination efficiency*). Therefore, we analyzed the angular amplitude of the forearm, wrist and index finger during the striking movement in order to characterize interindividual variability in motor coordination. For a first approximation, we looked for two motor strategies recruiting joints able to generate enough cube speed at impact, namely, via wrist and/or forearm rotation. It is important to note, here, that "motor strategy" is not intended to reflect a qualitative change in motor coordination in terms of generalized motor program or motor primitive, but rather a substantial quantitative difference in movement parameters.

We used a non-hierarchical K-mean clustering (K = 2) [41] on the joint angular amplitude values during the striking movement (for the elbow, wrist and index finger rotations) in the last trial of each condition, which we assumed reflected relative stability of the motion. Fourteen participants were identified in a "Wrist strategy" cluster, and nineteen in a "Forearm strategy" cluster. Fig 8 illustrates a typical trial for one participant in each cluster, for the 50 cm target distance on a Balsa wood surface with a 0˚ surface slope. This condition corresponds to one of the most demanding conditions in terms of optimal cube speed (see Table 2). The Wrist strategy shows greater angular amplitude and greater angular speed for the wrist rotation, as compared to the other segments. By contrast, the Forearm strategy shows recruitment of both the forearm and the wrist. This different involvement of the upper limb segments of interest in each strategy is also illustrated in Fig 9 where the angular amplitudes are plotted as a function of the optimal cube speed required to reach the target.

Under the hypothesis of linear recruitment of a given joint as a function of the increase in task demand, we retained linear fits with $R^2 > 0.75$ as significant. Similar to Figs 8 and 9 shows an equivalent recruitment of the wrist and forearm in the Forearm strategy and a much greater

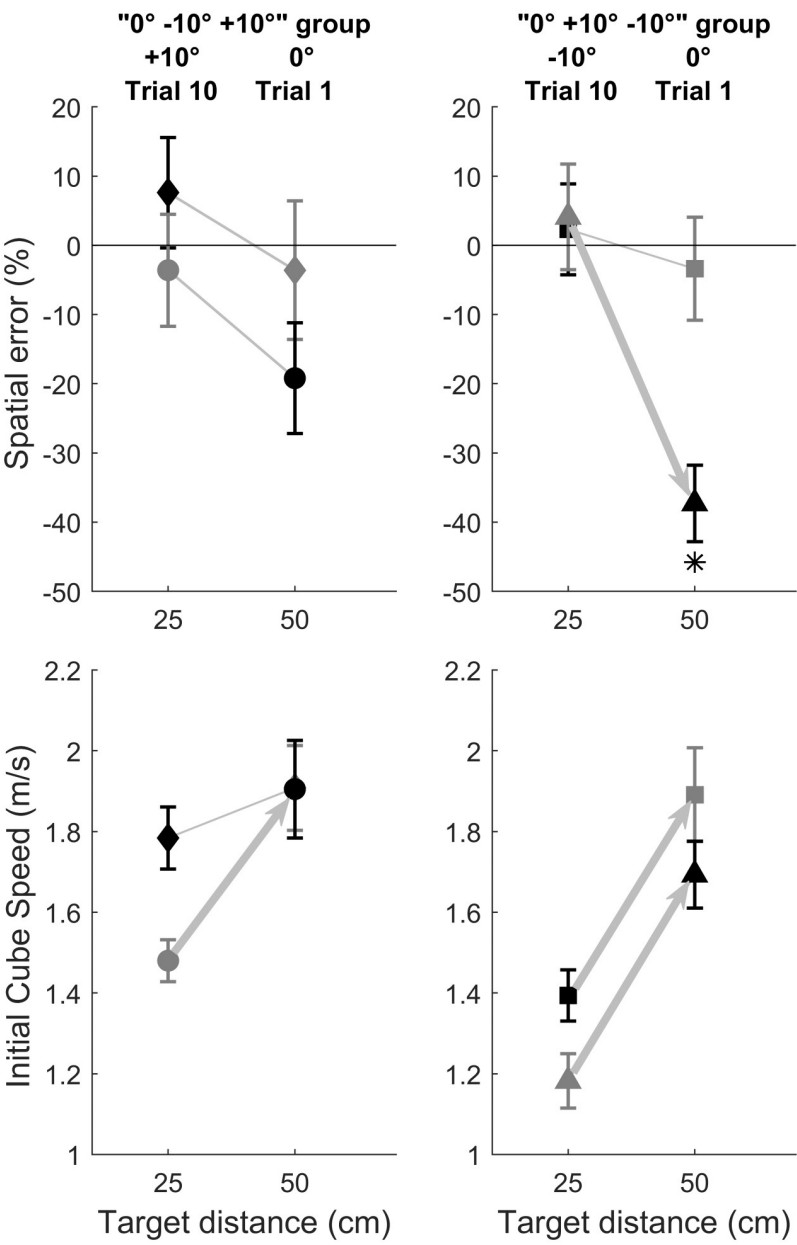

**Fig 7. Illustration of performance after a change in three physical properties of the stimulus, i.e., surface slope, surface material and target distance.** This change corresponded to the transition between the last trial of block 6 and the first trial of block 7 for the four different groups in this study (see Table 1 for details about block order). Each group is illustrated with a different data point symbol. Gray and black symbols indicate aluminum and balsa wood surface material, respectively. For example, diamonds correspond to a "0˚, -10˚, +10˚" group where the last trial of block 6 was a +10˚ slope with a balsa wood surface material and 25 cm target distance, followed by a 0˚ slope with an aluminum surface material and 50 cm target distance. Asterisks illustrate significant differences in spatial error with respect to zero (Bonferroni correction with α = 0.05/4), and arrows illustrate a significant change in performance between the trials of interest (Bonferroni correction with α = 0.05/4). The results show appropriate movement re-parameterization after a change in three physical properties of the stimulus, except for the more difficult transition indicated by the triangle data points corresponding to a transition from a 25 cm target distance on a -10˚ aluminum surface (optimal initial cube speed is 0.967 m/s, see Table 1) to 50 cm target distance on a 0˚ balsa wood surface (optimal initial cube speed is 2.147 m/s, see Table 1).

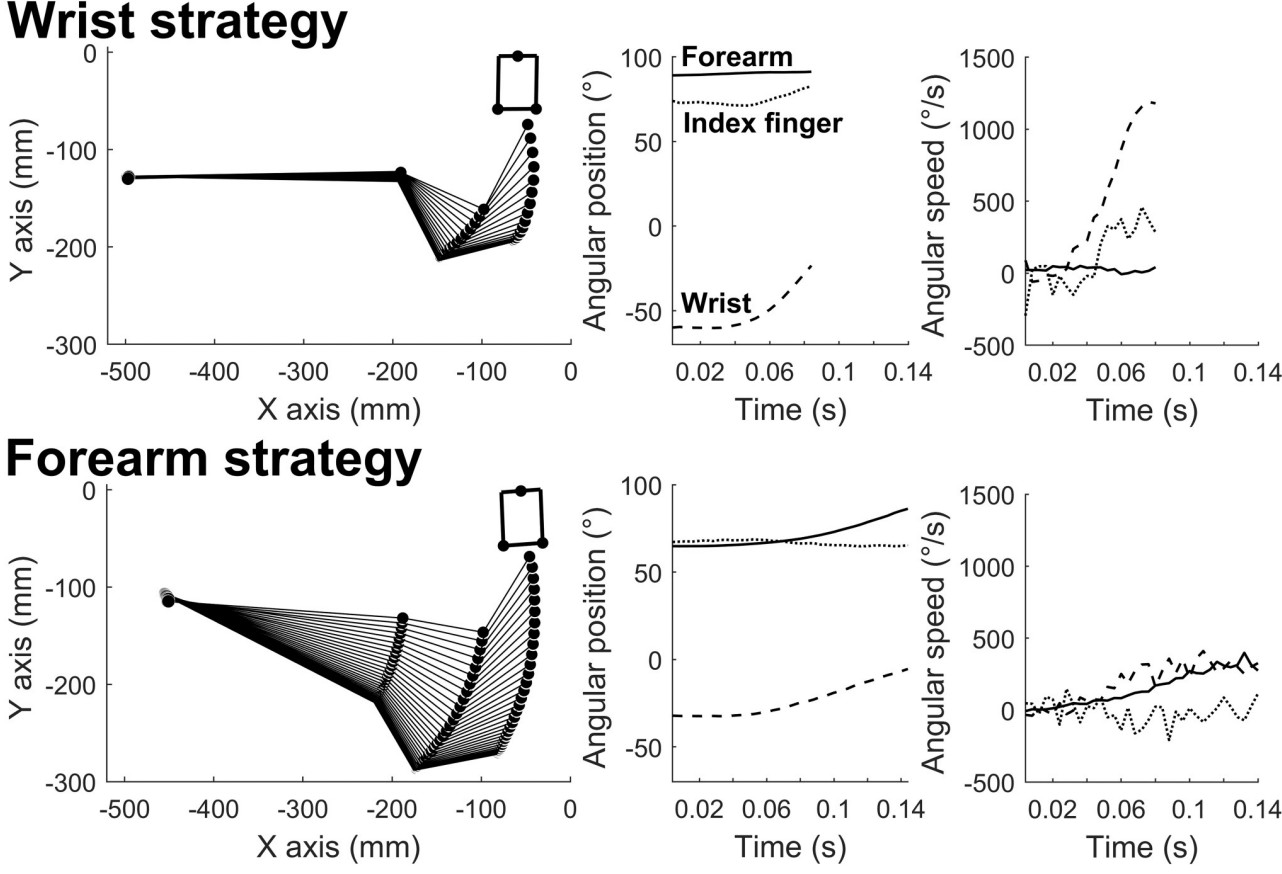

**Fig 8. Individual difference in motor coordination.** Kinogram representation of one striking movement, together with corresponding kinematic data, in the 0° slope condition, 50 cm target distance and balsa wood, for one participant in each group cluster. The "Wrist strategy" participant (upper panels) shows greater angular amplitude and greater angular speed for the wrist rotation, as compared to the other segments. By contrast, the "Forearm strategy" participant (lower panels) shows recruitment of both the forearm and the wrist.

recruitment of the wrist in the Wrist strategy. In order to further investigate the contribution of each limb segment, we conducted separate repeated measures ANOVA for each segment on the individual slope and intercept values, using Motor strategy as a between-subjects factor, and Surface material as a within-subject factor.

Regarding forearm rotation, each $R^2$ was greater than 0.75 (see Fig 9), and the ANOVAs showed significantly greater intercepts only for the Forearm strategy ($M = 5.02°$, CI = [1.21; 8.83]) compared to the Wrist strategy ($M = -1.21°$, CI = [-4.56; 2.13]), $F(1, 31) = 6.22$, $p = 0.018$, with a significant intercept (with respect to zero) only in the former group, $t(18) = 2.77$, $p = 0.013$. There was no other main effect or interaction on intercepts. Moreover, the ANOVA on slopes showed no main effect or interaction, although each of them differed from zero, all $ps < 0.02$. Overall, these results suggest a greater initial recruitment of forearm rotation for the Forearm strategy, and similar modulation of forearm amplitude in both groups with increasing task demand. The absence of Surface material effect within each group simply reflects the fact that parameterization of joint angular amplitude is based on optimal cube speed which varies as a function of the combination of the three physical properties (target distance, surface slope, and surface material).

For wrist rotation, given the $R^2 > 0.75$ criterion, we limited our analysis to the effect of Motor strategy in the Aluminum condition (see Fig 9). The intercept was significantly greater

# Forearm angle  # Wrist angle  # Index angle

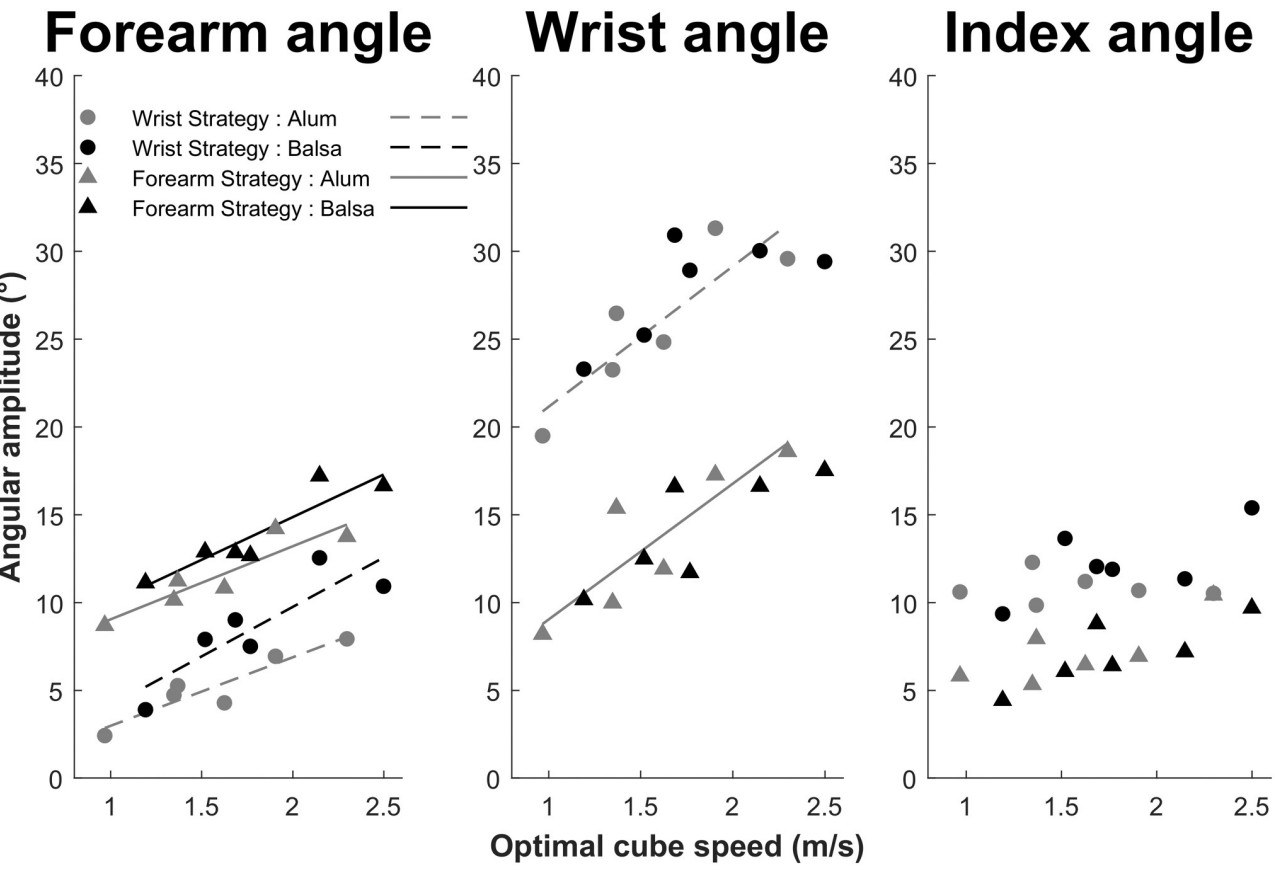

**Fig 9. Motor coordination parameterization as a function of the optimal cube speed reflecting the task physical constraints.** In the last (10th) trial, participants show a linear regulation (only $R^2 > 0.75$ are illustrated) of forearm angular amplitude for each surface material, and of the wrist angular amplitude only for the aluminum surface. Participants adopting a Wrist strategy show a greater intercept in wrist angle regulation.

for the Wrist strategy ($M = 13.15°$, CI = [8.01; 24.77]) compared to the Forearm strategy ($M = 1.30°$, CI = [1.32; 6.89]), $t(31) = 2.37$, $p = 0.024$, with a significant intercept only in the former group, $t(13) = 4.22$, $p = 0.001$. By contrast, the slope for the Wrist strategy group ($M = 8.00°/m.s^{-1}$, CI = [3.32; 12.68]) and the Forearm strategy group ($M = 7.73°/m.s^{-1}$, CI = [3.34; 12.12]) did not differ significantly, $t(31) = 0.09$, $p = 0.93$, but they both differed from zero, all $p$s < 0.003. These results are consistent with a greater initial recruitment of wrist rotation for the Wrist strategy. The nonlinear recruitment of the wrist in the Balsa wood condition is possibly due to a trade-off between the recruitment of wrist and finger joints for the more demanding conditions. Indeed, wrist angle caps after 2 m/s optimal cube speed, while finger angle increases more than for the less demanding conditions.

Although we found no linear trend (each $R^2 < 0.75$) for finger angular amplitude as a function of optimal cube speed, we further investigated the motor control of the striking movement by examining how the index fingertip speed at impact was regulated as a function of optimal cube speed. The results showed that both variables were linearly related, as illustrated in Fig 10. The ANOVA on intercept values evidenced a significantly greater intercept for the Wrist strategy ($M = 1.16$ m/s, CI = [0.80; 1.52]) than for the Forearm strategy ($M = 0.61$ m/s, CI = [0.38; 0.85]), $F(1,31) = 8.06$, $p = 0.008$. In addition, the intercept for Balsa wood ($M = 1.43$ m/s, CI = [0.94; 1.91]) was significantly greater than for Aluminum ($M = 0.90$ m/s, CI = [0.52; 1.28]), $F(1, 31) = 8.19$, p = 0.007, reflecting the fact that, with balsa wood, greater finger speed

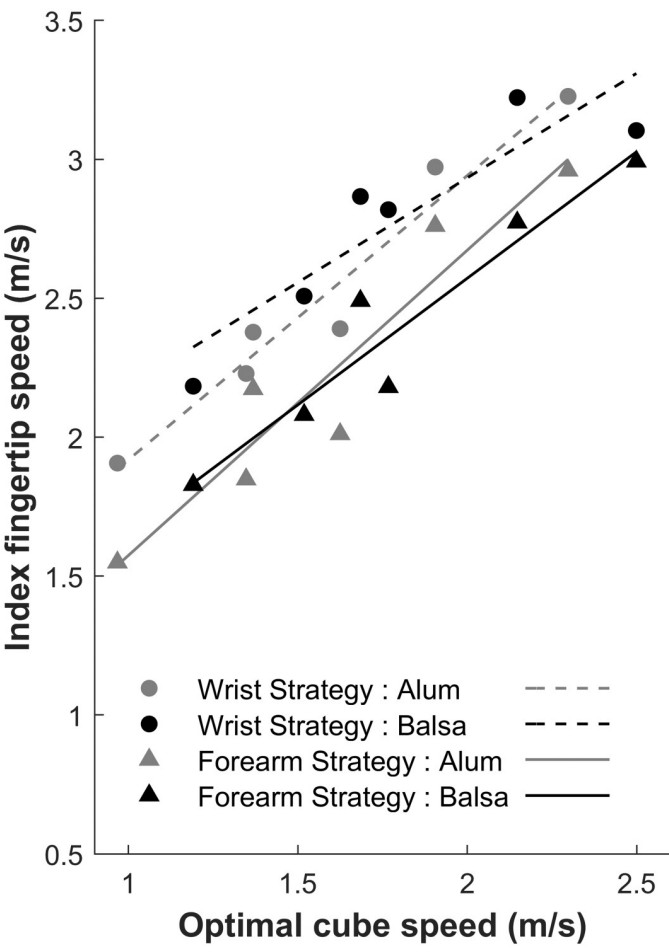

**Fig 10. Individual variability in movement parameterization.** The two motor strategies are also reflected (in the last trial) at the distal level through the linear regulation of index fingertip speed at impact as a function of optimal cube speed. In order to reach the required amount of KE at impact, participants adopting a Forearm strategy reduced overall finger speed as compared to the other strategy, due to greater limb mass put into motion.

is needed at impact to counteract the effect of greater friction on cube displacement. Neither factor interacts on finger speed at impact intercepts, $F(1, 31) = 0.84$, $p = 0.37$, n.s. The ANOVA on slopes showed a significant effect only of Surface material, $F(1,31) = 8.67$, $p = 0.006$, with a greater slope for Aluminum ($M = 1.02$ m/s, CI = [0.76; 1.29]) compared to Balsa wood ($M = 0.75$ m/s, CI = [0.49; 1.01]). These results suggest that the greater friction with Balsa wood impedes more fine-grained movement regulation (smaller slope corresponds to smaller variation in finger speed).

## Motor coordination efficiency

In order to investigate differences in behavior related to motor coordination, we first examined if mean spatial error and cube speed at impact varied with Motor coordination (wrist vs. Forearm strategy). The groups did not differ, either in terms of mean spatial error (Wrist strategy $M = -4.45\%$, SE = 2.47; Forearm strategy $M = -0.02\%$, SE = 2.18, $t(31) = 1.33$, $p = 0.19$), or in terms of initial cube speed after impact (Wrist strategy $M = 1.68$ m/s, SE = 0.03; Forearm strategy $M = 1.74$ m/s, SE = 0.02, $t(31) = 1.66$, $p = 0.11$). However, the index fingertip speed at

impact was significantly greater in the Wrist strategy group ($M$ = 2.62 m/s, SE = 0.11) than in the Forearm strategy group ($M$ = 2.25 m/s, SE = 0.07), $t(31)$ = 3.20, $p$ = 0.003.

In order to account for similar (spatial error) performance in spite of different index fingertip speed at impact, we investigated upper limb total kinetic energy at impact (Total KE). This is the sum of the forearm and hand KE (see *Materials and Methods* section for details on computation). We assumed that Total KE is the physical parameter that must be controlled by the brain during the striking movement in order to send the cube to the target distance corresponding to a given optimal cube speed after impact. In turn, the latter is the kinematic consequence of the cube KE transmitted by fingertip-cube impact.

Fig 11 illustrates the relationship between the kinetic variables (of the upper limb and the cube) and the kinematic cube speed variable at impact corresponding to the optimal cube speed values of all the experimental conditions. To reach a specific target distance, a particular amount of Total KE is needed, whatever the motor strategy. The quality of the linear fits suggests that the Forearm strategy participants ($0.96 \leq R^2 \leq 0.99$) better regulated upper limb KE for striking movements at the tenth trial than the Wrist strategy group did ($0.79 \leq R^2 \leq 0.92$). However, regarding the upper limb KE values, the ANOVAs on slopes and intercepts showed no main effect of group or Surface material, nor any interaction between the two factors (all $p$s > 0.16). By contrast, the cube KE showed marginally greater slope values for the Forearm

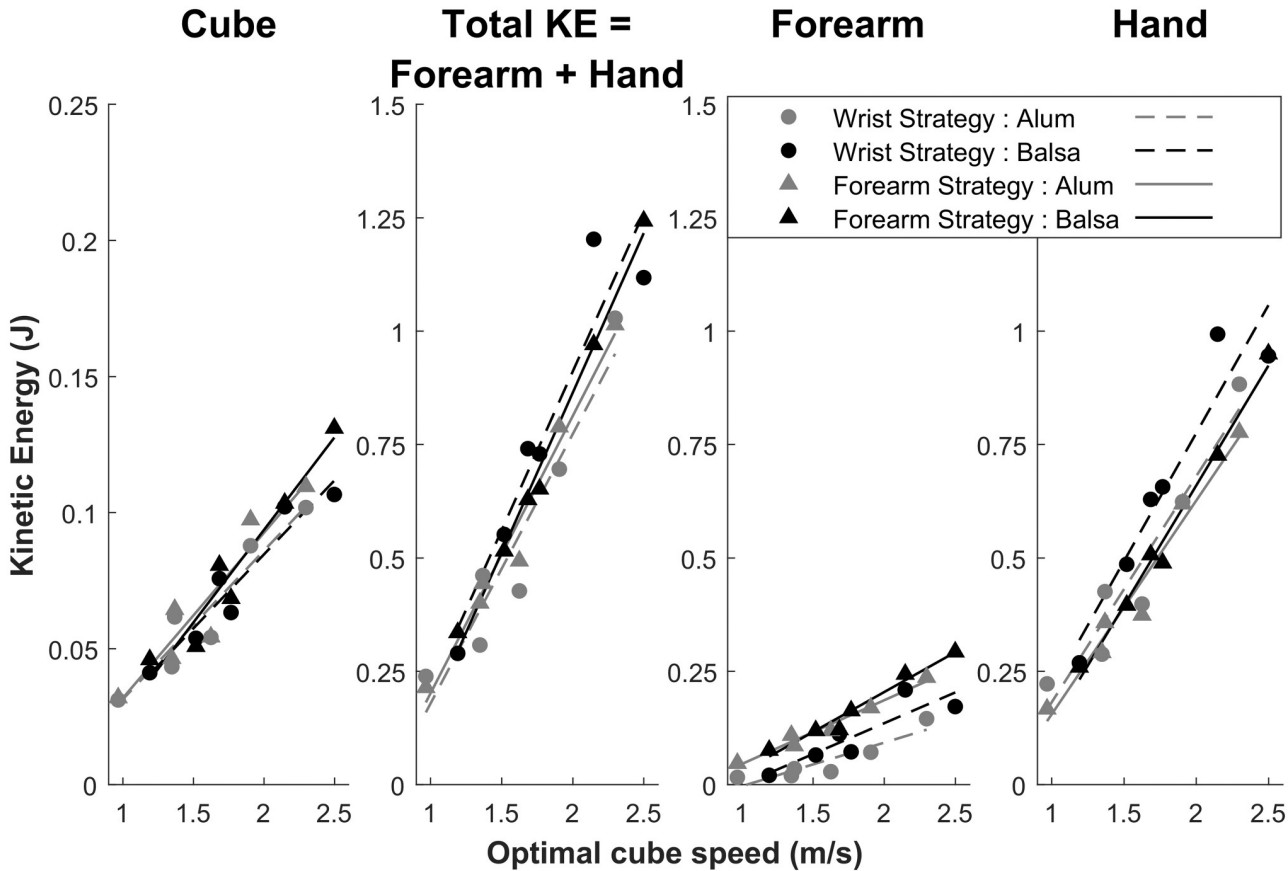

**Fig 11. Motor coordination efficiency is determined by kinetic energy at impact.** This figure illustrates (for the last trial) that movement regulation is linearly determined by the amount of kinetic energy at impact needed to impart on the cube the optimal speed corresponding to the different task constraints.

strategy ($M$ = 0.064 J, CI = [0.058; 0.071]) compared to the Wrist strategy ($M$ = 0.055 J, CI = [0.047; 0.062]), $F(1,31)$ = 4.08, $p$ = 0.052, with no effect of Surface material, nor any interaction between the factors. In addition, the intercepts did not vary (all $p$s > 0.12). The greater cube KE slope for the Forearm strategy as compared to the Wrist strategy (together with similar intercept) reflects the fact that more KE is provided by the greater limb mass in motion, especially with increasing optimal cube speed. Moreover, this result echoes the descriptively closer-to-optimal spatial error performance of the Forearm strategy ($M$ = -0.02%) compared to the Wrist strategy ($M$ = -4.45%); see above. Finally, mean Forearm KE was significantly greater for the Forearm strategy ($M$ = 0.15 J) than for the Wrist strategy ($M$ = 0.08 J), $t(31)$ = 2.64, $p$ = 0.013, although the groups did not differ in terms of Hand KE and Total KE (both $p$s > 0.11). This finding suggests that due to the greater Forearm KE in the Forearm strategy group, participants reduced index fingertip speed at impact (as compared to the Wrist strategy) in order to reach equivalent Total KE in both groups. However, although Total KE linearly predicts Cube KE (Wrist strategy $R^2$ = 0.92; Forearm strategy $R^2$ = 0.96), the slope of Cube KE as a function of Total KE is far from 1 (Slope$_{\text{Wrist strategy}}$ = 0.075; Slope$_{\text{Forearm strategy}}$ = 0.097), reflecting the fact that KE efficiency (Cube KE/Total KE) is only about 10%. The other 90% is conserved in the arm movement after contact and in the deformation of the finger(tip) during contact.

## Discussion

When sliding an object along a surface toward a target distance, initial object speed after impact is the crucial physical parameter. This optimal speed depends on the distance to be reached, the coefficient of friction μ (which depends on the material of both the object and the surface on which it moves), as well as the inclination of the surface (see Eq 3). Here, we wanted to determine whether the brain uses initial cube speed as a kinematic control variable in order to control arm movement parametrically to transmit the appropriate KE to the object. We also examined how upper limb segment motor coordination is regulated depending on interindividual variability.

The literature shows that we tend to underestimate μ when judging a surface material visually [42]. Likewise, our data suggest that underestimation of μ corresponds to a default estimate for horizontal surfaces, as illustrated by the initial spatial error showing systematic undershoots in our study. Indeed, spatial error somehow directly reflects participants' internal model of μ (subjective μ). To demonstrate that, let's change Eq 3 in order to find μ as a function of cube speed at impact (Cube Speed$_{\text{impact}}$):

$$\mu = \frac{\left(\frac{\text{Cube Speed}_{\text{impact}}^2}{2\,d\,\text{g}}\right) - \sin(\alpha)}{\cos(\alpha)} \quad (9)$$

If we were to retrieve the *subjective μ* from the data, at a given trial, we would use the target distance $d$ together with the observed Cube Speed$_{\text{impact}}$ because the participant imparts a given cube speed at impact thinking that the target distance will be reached. By contrast, the *objective μ* value would be computed using the actual distance traveled by the cube for the given trial. As a consequence, *subjective μ* would be a ratio between the actual $d$ value (traveled by the cube) and the target distance. This ratio is actually reflected in the spatial error. Of course, our reasoning builds on the assumption that we have a "pretty good" internal model of gravity [30]. If participants' internal model of gravity is wrong, then the brain would need to solve Eq 9 with two unknown factors, namely *μ* and *g*. However, motor control would still be possible provided the subjective *g* value is a constant. If the subjective *g* value is greater than 9.81 m/s$^2$

then the subjective $\mu$ would be smaller than the actual $\mu$ value. The brain would only need to apply a scale value to the subjective μ in order to parameterize the movement so that it provides the required initial cube speed.

Moreover, we found that, at the first trial in the initial 0° slope condition, the participants imparted a similar initial speed to the cube regardless of the Surface material, which in turn led to greater undershoots for the 0° surface material with greater μ (see Fig 3), namely, balsa wood. We hypothesize that keeping cube speed constant facilitated the estimation of μ (from the visual consequences of the cube kinematics), and the subsequent parameterization of the movement in order to reduce spatial error across subsequent trials. This parameterization is reflected in the adaptation curves aiming at increasing initial cube speed to reduce undershooting (see Fig 3 for the 0° condition). As a consequence, after ten trials, the brain may have converged toward a first approximation of μ. Data for the -10° condition follow these lines, showing pretty good performance with no signs of adaptation (see Fig 3). Moreover, further evidence of the calibration of *subjective μ* (internal model of friction coefficient) is provided by the significant decrease of mean cube speed at the initial -10° trial compared to the final 0° trial.

Although we agree with Hadjiosif et al. (2021) [43] that motor adaptation within block does not require updating of an internal model, our results provide some evidence that when a physical property, such as surface slope, is changed between blocks, participants can appropriately re-parameterize their movement to directly reach optimal performance. This adapted behavior cannot be explained without invoking an internal model of physics, or luck. On the one hand, the fact that performance is sometimes not optimal after block change suggests that updating a predictive forward model for movement control is done in parallel with, and independently of, the mechanisms underlying error-based movement adjustment across trials of a given block (see the so-called "direct policy learning" of Hadjiosif et al., 2021[43]). On the other hand, participants sometimes showed directly optimal performance at the first trial of a block despite a dramatic change in physical properties such as surface slope, surface material, and target distance simultaneously (see Fig 7). This suggests that the adequate change in cube initial speed at the initial trial of this new block resulted from a predictive forward model-based movement re-parameterization. This was possible because participants tuned their subjective μ of the surface material during previous blocks with a different target distance (see Table 1, the first three blocks with the same surface).

The only case for unsuccessful movement re-parameterization when three physical properties changed between blocks corresponded to the more demanding transition (see Fig 7) that required an increase of about 120% of initial cube speed (see Table 2, from 0.967 m/s for a 25 cm target distance on a -10° aluminum surface, to 2.147 m/s for a 50 cm target distance on a 0° balsa wood surface). In this case, participants undershot the target distance. If one considers that predictive forward models are encapsulated knowledge of the physical properties of the environment and of the motor system [44], underestimation may reflect the influence of explicit/conscious knowledge on motor control. This explicit influence may correspond to adaptive prior knowledge [45] according to which underestimation is relevant for survival. Along the same lines, Joh et al. (2007) [12] showed that underestimating high friction slopes permitted participants to attempt standing on steep slopes without risk of slipping.

The main challenge for the brain was movement control under greater task demands. The participants used a linear regulation of cube KE as a function of optimal cube speed that would lead to optimal performance (see Fig 11). This regulation of cube KE somehow reflected the linear regulation of forearm and wrist (to a lesser extent) angular amplitudes as a function of optimal cube speed (see Fig 9). This control of limb segment angular amplitude fits well with the literature on postural arm control [46]. Following feedback error learning models [28,29],

we propose that spatial error provides a sensory error signal that is used to regulate arm posture and KE in a predictive way (in order to impart the required cube speed) via feedforward motor commands [47]. The parietal cortex and cerebellum would be the core brain structures for this perception-action regulation [48] and the motor adaptation that we evidenced across trials. Spatial error (measured by parietal regions) would provide a control variable to the brain in order to converge towards the actual *objective μ* value needed to parameterize the striking movement. However, at the distal level, fingertip kinematics also reflected interindividual variability in the limb mass put into motion (see Fig 10).

In this study, we examined interindividual variability in striking movement parameterization by analyzing the angular amplitude centered on the forearm, wrist and index joints. The idea is that the brain takes advantage of available motor redundancy [31,32] to cope with the increasing task demand characterized by a continuum of optimal cube speed just after impact depending on the combination of surface slope, target distance and surface material, in order to reach a given target distance. The linear trends regressing motor behavior (in terms of angular amplitude, fingertip speed and KE) on optimal cube speed provided evidence in favor of such a continuum in movement parameterization. However, in order to characterize interindividual variability in motor coordination, we chose a cluster analysis on angular amplitude at the joints of interest for the different conditions. Analysis of these angular amplitudes showed a major contribution of the forearm and hand segments to the striking movement, which makes sense given that, unlike the finger, those limb segments provide sufficient mass to reach the required KE values that would lead to optimal cube speed in our experiment. Therefore, we limited our analysis to two clusters of participants that would show a substantial quantitative difference in movement parameters.

To strike an object which then slides toward a target distance, a specific amount of kinetic energy must be transmitted during impact. Although both clusters of participants showed a linear increase in the KE variables (of the upper limb and the cube) with optimal cube speed, they differed at the behavioral kinematic level. The Forearm strategy group was characterized by a similar variation of angular amplitude of the forearm and wrist with optimal cube speed. On the other hand, the Wrist strategy group showed much higher values for the wrist and lower values for the forearm than the other group. This group variability in limb segment angular amplitudes was paralleled by a difference in fingertip speed at impact. In order to reach similar levels of cube KE as a function of optimal cube speed, participants using a Wrist strategy increased the rotational KE of the hand to compensate for their lower Forearm KE (forearm mass in motion) compared to the Forearm strategy. This increased rotational KE of the hand in the Wrist strategy was reflected in greater wrist angular amplitude and led to greater finger speed at impact (see Figs 8 and 10) compared to the other group. However, because forearm mass translation enters in the Hand KE formula (see Eq 6), the groups did not differ in terms of Hand KE. In line with studies showing that the motor cortex may represent arm movement through intrinsic parameter spaces of joint kinematics and joint torques [49], our results suggest that the KE of upper limb segments may constitute a controlled parameter in the parameterization of the striking movement. Moreover, our group differences suggest that motor redundancy allows for individual variability in movement parameterization through underlying muscle synergies. KE would provide an optimality criterion for the brain in the compositional construction of the movement supported by force-field/muscle synergy motor primitives, along the lines of Giszter [50].

Our findings on upper limb KE control by the brain echo the literature on stone knapping with a hammer. It has been shown that when hammers with different mass to fractionate stone are used to obtain a similar sized flaking stone, participants control the specific amount of kinetic energy to obtain the same performance [23]. Together with our findings, this literature

suggests that the brain takes kinetic variables into account to parameterize movement, whether from the body limb or from the environment with which we interact. Our study complements others showing that the brain has a "pretty good" (i.e., functionally relevant, although not perfect) internal model of gravity [30] by providing evidence of an internal model of friction for the motor control of striking movement for the purpose of sliding an object towards a target distance. However, further research is needed to address certain limits of our work. Among others, the spatio-temporal regulation of the striking movement should be examined in greater detail to determine which variable is regulated (movement duration while keeping angular amplitude constant or movement amplitude together with an invariant movement duration) to transmit KE to the object. Likewise, along the lines of Sternad (2018) [31], we suggest that variability in spatial error across trials, and of KE as a function of optimal cube speed, may reflect random exploration of motor strategies in order to achieve the task. The source of this "motor noise" is intentional (for the purpose of reducing spatial error), contrary to physiological noise in neural control signals (Harris & Wolpert, 1998) [51] that may also add variance in performance. Participants intentionally learn to control various sources of noise in performance such as KE efficiency (the loss of KE between Total KE and Cube KE), the orientation of cube trajectory after impact (to avoid or exploit collisions between the cube and the gutter), etc. These different sources of noise may also account for the partial generalization of subjective μ between trial blocks.

Similarly, further investigation of the role of visual perception of initial object cube speed for the control of movement in our task may require experimental manipulation of both visual input and object mass. As illustrated in Eq 10, object mass has no impact on optimal object speed to reach a given target distance (see the left part of the equation). What is important is the initial object cube speed irrespectively of its mass. Let's take an example with a cube sliding on a horizontal surface where the participant performs a block of trials on an aluminum surface, and then another block on a balsa wood surface. If the subjective μ value of each surface has been correctly tuned at the end of each block, then if the participant is given a cube of different mass (a preliminary psychophysical experiment may be needed to estimate the participant's differential threshold for object mass), we expect that he/she will send the new cube to the target distance accurately, imparting the appropriate cube KE to reach the same initial cube speed as before for each surface material. By contrast, if the first half of the target distance was occluded in the initial two blocks, so the initial cube speed was never available visually, we expect substantial spatial error after the change in cube mass.

$$\text{d} = \frac{1}{2} * \frac{v_0^2}{g[\sin(\alpha) + \mu\cos(\alpha)]} = \frac{1}{2} * \frac{v_0^2 * mass}{(g\,[\sin(\alpha) + \mu\cos(\alpha)] * mass)} = \frac{KE}{(resistive\,force)} \quad (10)$$

Finally, neurophysiological variables, already used to establish the internal model of gravity, such as EMG on upper limb muscles [1], could also be applied to determine how the internal model of friction modulates movement parameterization by modeling time-varying muscular synergies [52].

## Author Contributions

**Conceptualization:** Sylvain Famié, Mehdi Ammi, Vincent Bourdin, Michel-Ange Amorim.

**Data curation:** Sylvain Famié.

**Formal analysis:** Sylvain Famié, Vincent Bourdin, Michel-Ange Amorim.

**Investigation:** Sylvain Famié, Michel-Ange Amorim.

**Methodology:** Sylvain Famié, Mehdi Ammi, Vincent Bourdin, Michel-Ange Amorim.

**Supervision:** Mehdi Ammi, Michel-Ange Amorim.

**Writing – original draft:** Sylvain Famié, Michel-Ange Amorim.

**Writing – review & editing:** Sylvain Famié, Michel-Ange Amorim.

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
