## [Decision Letter · Decision Letter 0]

7 Jun 2021

PONE-D-21-08226

Evidence for an internal model of friction when controlling kinetic energy at impact to slide an object along a surface toward a target

PLOS ONE

Dear Dr. Famié,

Thank you for submitting your manuscript to PLOS ONE. After careful consideration, we feel that it has merit but does not fully meet PLOS ONE’s publication criteria as it currently stands. Therefore, we invite you to submit a revised version of the manuscript that addresses the points raised during the review process.

Please make sure to carefully answer all of the reviewers' concerns, specifically those of two of the reviewers questioning the validity of your conclusions based on the data you are presenting. As you can see, reviewer 1 doubts that your study provides evidence for an internal model of friction, and reviewer 3 also has concerns. 

All three reviewers and myself noted that data and code are not available at the link provided. Please make sure that this is the case.

We look forward to receiving your revised manuscript.

Kind regards,

Stefan Glasauer

Academic Editor

PLOS ONE

Journal Requirements:

2. Thank you for including your ethics statement:  "The experiment was approved by the local ethics committee (CER-Paris-Saclay-2018-021-R).".   

Reviewers' comments:

Reviewer's Responses to Questions

**Comments to the Author**

1. Is the manuscript technically sound, and do the data support the conclusions?

Reviewer #1: No

Reviewer #2: Yes

Reviewer #3: Partly

2. Has the statistical analysis been performed appropriately and rigorously? 

Reviewer #1: Yes

Reviewer #2: Yes

Reviewer #3: Yes

3. Have the authors made all data underlying the findings in their manuscript fully available?

Reviewer #1: No

Reviewer #2: No

Reviewer #3: No

4. Is the manuscript presented in an intelligible fashion and written in standard English?

Reviewer #1: Yes

Reviewer #2: Yes

Reviewer #3: Yes

5. Review Comments to the Author

Reviewer #1: The main claim of this study is that it has obtained experimental evidence that human motor control system uses internal model of friction. There is plenty of evidence for internal models of friction and clever strategies how they are used; regrettably the authors mention only a very limited set of studies. The novelty in this study is claimed to be that it investigates ballistic movements.

Unfortunately the study design is flawed and there is absolutely no evidence of internal model of friction demonstrated in this study. By classical definition of "internal model" subject should be able to parameterize movements based on knowledge of friction. Ability to learn to perform movement when friction changes, is not an evidence of internal model of friction.

In this study the subjects' task was to hit a plastic cube-like object for it to slide to a given target distance. The subjects learned to perform this task minimizing the distance error. Then the same task was performed during 10 trials when surface was sloped at 10 degrees followed by another 10 trials when surface was sloped by 10 degrees in the opposite direction. After that the surface material and thus friction was changed and the procedure repeated. Again the subjects learned to hit the cube with appropriate speed and transfer kinetic energy (KE) to make it slide closer to the target. I see no discovery in the fact that people seeing under/over-shoot in the target distance can figure out that they have to hit stronger or weaker. For this to happen no model of friction is needed. Put it this way – if there would be no internal model of friction, what would the result be then?

Even with this flawed experimental design one might attempt to extract some hints of possible internal model of friction existence, if learning speed would be compared between the first block of conditions and the second when surface friction was changed. No such analyses was attempted and data were analysed regardless whether given frictional condition was present in the first or in the second block.

The conclusion that " internal model of friction can be generalized when the slope changes" is simply wrong and the logic is upside down what the experimental design allows to conclude - the friction was not changed and remained the same when blocks with different slope angles were tested. How authors arrived at such conclusion is absolutely not clear. If anything, this study reconfirms that there might be an internal model for slope angle, but absolutely no evidence for internal model of friction.

For example, the simplest experimental design, which actually would be able to demonstrate internal model of friction, might be that the subjects would learn to slide the same cube within two channels, lined with two surfaces with different frictional properties. Then another cube with a different mass would be introduced. Subject would learn to slide this cube to hit the target distance within one of the channels with one of surfaces. Then taking this cube to a second channel with a different, but known friction and repeating the task. If there would be internal model of friction then subjects would be able to apply the friction model to adjust KE to the new frictional condition. This would be evidence of frictional model as no explicit learning of KE would be possible as no such condition has been encountered before. Thus it would indicate that the required KE must have been estimated (modeled) based on internal model of friction coefficient.

The second largest section of the manuscript deals with multiple strategies how subjects regulated KE of the hit in this constrained experimental situation. I don’t see how it is directly relevant to the question about internal model of friction, thus I see no value in it, but I hope this information could be useful for someone. The authors would need to explain better what it reveals what is important beyond specifics of this manuscript.

Minor

The directory address, where the authors claim the study data are made available, was empty at the time of review.

Reviewer #2: General: The manuscript by Famie and colleagues reports a study which aimed to provide evidence for the existence of a context-specific representation of the friction of a surface, over which an object had to be pushed. Participants were instructed to slide with a hitting motion an object across surfaces of varying friction (aluminum vs. balsa wood) at a specific target distance (25 vs. 50 cm) under different conditions of surface slope (0, 10, -10 degrees). For each task condition, performance curves were plotted for the spatial error of reaching the target distance and the initial speed of the object after it was hit. The experimental protocol was arranged in such a manner that participants were first exposed to the horizontal surface to acquire an estimate of the friction of each surface. Learning was demonstrated by fitting exponential performance curves across the ten trials of each condition. In terms of a transfer condition, participants than had to perform in the up- or downward sloped conditions, which did not express similar learning but nearly optimized performance instead. Based on the motion equation of the observed initial speeds, the participants’ subjective friction estimates were derived. The progression of the spatial error of reaching the required distance was considered the indication of the acquisition of the internal model of a surface’s friction. In addition, participants’ arm and wrist kinematics during the striking motion were analysed to determine their individual strategies for regulating to invested amount of kinetic energy. Two groups of participants could be established: one group that only used a wrist motion and one group that used an elbow and wrist motion.

The study described in this manuscript made use of a very sophisticated methodology. I appreciate its scientific rigor and interesting conclusions. I follow the authors’ arguments but the only aspect that I miss in the discussion of the manuscript is an integrated perspective, which combines an internal model of friction with an internal model of gravity. Do the authors believe that a representation of friction is the only factor that influences performance in the two conditions, where the surface shows a downward or upward slope, or how are the two representations combined? Perhaps the author can elaborate this a bit more.

Minor:

Page 3, paragraph 1: “Amorim […] where an object […] should stop after being launched due to a collision (with a launching object)”. I find this summary of the Amorim et al. study unclear, perhaps it can be clarified better.

Page 9, equation 3: the equation lacks a squared bracket

Page 9, table 1: I find the ordering of the table counterintuitive

Page 11, paragraph 2: “We used […]”. This sentence is redundant given the first sentence of the first paragraph of page 10.

Page 11, paragraph 3: “Motor adaptation […]”. Where exponential curves fitted for each individual participant or just for the group averages? I think the latter would be a weakness as it is unclear how well the function fits represent the individual participants performance curves.

Page 14, paragraph 2: “The ANOVA also showed […]”. Which ANOVA? Perhaps here it could be a bit more specific.

Page 15, paragraph 1: This description of Figure 4 does not seem to follow the order of the figure panels. Perhaps the description could refer to the figure panels better.

Reviewer #3: Review of PONE-D-2108226

Summary

The authors examine whether humans have and update an internal model of friction. To this end, they asked participants to slide objects of different materials to two different distances along a path that had one of three different inclinations. By presenting conditions in a blocked manner, the authors could examine whether motor behavior developed over trials on the basis of the observed error between the required target position and the cube’s final position. The results show that participants adjusted their behavior over the trials to foster proper task execution, but only in some conditions. Despite the adjustments in some of these conditions, spatial error in the last trial was sometimes pronounced. These suggest that humans can utilize the observed outcome of a performed trial to update the motor plan of the upcoming trial, however this motor updating seems to occur under specific circumstances and appears incomplete.

First comments

This novel study will be of interest for the scientific community, especially those disciplines related to the planning and control of human movement and the role of physical properties, such as friction, in this. The main claims are incorporated within appropriate theoretical frameworks. The analyses are generally suitable. The results support some of the claims, though I have some suggestions below for improvements in this aspect. For instance, a general calibration of the internal model of friction, as implied in the abstract, cannot be concluded based on the data. Generally, the presentation of the results, and as a consequence the appreciation of the conclusions, could be more comprehensible.

The data and experimental code are not available right now. The authors provide a link where some material can be found, but the linked folder is empty.

Below I provide detailed comments on what I believe could be improved in the manuscript. I hope that the comments can be helpful to the authors to further improve their nice work.

Comments

One of the main claims is that an internal model of friction can be generalized when the slope of the sliding surface changes. This is a rather central claim, considering that it is also highlighted in the abstract of the study (lines 27-28). However, based on the results, this claim seems inappropriate. In fact, participants’ behavior for both objects is stably suboptimal across trials when the sliding surface is at -10 deg, as we can see in Figure 3 (upper left panel). Similarly, when the surface is at +10 deg, performance for the wooden object also does not improve across trials. These suggest that, even though participants can see that their actions did not bring the object at the correct place, they do not tailor their motor execution to the appropriate extent in order to overcome this error. As a matter of fact, the initial cube speed (Figure 3 – lower panels) is rather constant in the above-mentioned conditions. In contrast to this suboptimal behavior, participants do adjust their movements when the sliding surface is at 0 deg, for both objects, as well as for the aluminum-object when the sliding surface is at +10 deg. Nevertheless, such motor adjustments can be incomplete (wooden object at +10 deg), so the updating of an internal model may either be partial or the observed errors may arise from other sources than only the internal model (e.g., motor noise). These issues may deserve some more (explicit) discussion. In any case, I think it is important to avoid generalizations when it comes to the adaptation of the internal model.

A question that arises is why the adjustments are observed mainly in the 0 deg condition. It might be the case that participants have difficulties to estimate the role of gravity, which does not necessarily interact with friction. This too seems to deserve a more clear discussion.

Equation in line 183: It would be helpful to mention what are the variables in the equation (e.g., what is alpha?). Similarly for the equations in lines 223-224 (what is L?). Apologies if these are mentioned and I did not notice.

Equation in lines 223-224: There is a mistake in the bracketing, please adjust accordingly.

Equation in lines 223-224: Is it indeed true that the length of wrist is multiplied with the length of the elbow as seems to be the case in the end of line 223?

The presentation of the results sometimes becomes too technical and it is hard to get the main message. This is a general comment but particularly for lines 376-405. I recommend adding some higher-order information to guide the reader along the results and to highlight what do the reported effects mean.

Line 291: Why is the first trial of the +10 deg compared against the last trial of the 0 deg condition, and not with the last trial of the -10 deg condition, which is the condition presented just before. This way, the authors can examine whether the performance in trial_1 of +10 deg condition is similar to the most recently performed trial of the -10 deg condition.

The answer to this may be the explanation that is given in lines 259-261, however, in this case, this information is somewhat lost until the results are presented. Therefore, if this is the answer to my question, I suggest reminding the reader about the purpose and meaning of the comparison by giving this higher-order information at the respective Results-sentence. In any case, I am not fully convinced about the relevancy of this comparison (i.e. first trial in +/- 10 deg condition against the last trial of the 0 deg condition). If the purpose of the study is to examine whether participants tuned their internal model across repetitions of the same condition, then the authors should look whether participants learned from their error during the 10 trials of that conditions. What is the contribution to the research question of the comparison between trial_1 in +/- 10 deg and trial_10 of 0 deg condition? It just adds more statistical tests, so it makes it more likely to obtain a statistically significant result, which may in fact be random. Therefore, I would be tempted to suggest to remove any comparisons of the first trial in +/- 10 deg condition with the last trial of the 0 deg condition -unless it becomes explicit that such comparisons are central to the purpose of the study. Removing these would also help focus the results in the main, central findings, contributing to a more concise Results section.

Lines 446-448: The reduced finger speed at the time of contact in the Forearm-strategy group may indeed be due to a tuning of motor behavior on the basis of the increased kinetic energy, compared to the Wrist-strategy group. However, when rotating the wrist, the finger is by definition also rotated, so it is not surprising that the Wrist-strategy group shows higher index finger speeds. Considering this, the sentence in these lines may be somewhat misleading and it may require some rephrasing to also account for the above-mentioned possibility.

Minor

It would be helpful to add p-values in all reported tests, even if these are non-significant. This is sometimes done (e.g., line 281), but some other times it is not (e.g., 267).

Line 265: “significant spatial error”. What is the comparison that is being made here for the first trial “regardless of material”? Is it trial_1 vs some other trial? Or is it trial_1 between different orientations of the surfaces? Also, in line 266 there is a t-test result, but it is not clear what comparison this test refers to.

Line 291: “increased”. This reads as if there is an effect, however the statistical comparison does not support this wording. I recommend rephrasing.

Line 292: “in order to”…this reads as if participants aimed to achieve a spatial error. Consider rephrasing to something like “resulting in a similar spatial error…”.

Line 293: “was required”…so for the wooden object there was no requirement for motor adaptation? It is clear that already from the first trial there were spatial errors, and thus performance suboptimal. So, I would say, motor adaptation was required for both objects, but it was achieved only for the aluminum one. Still, the adaptation achieved in the aluminum object seems to be suboptimal as the spatial error appears quite strong in the last trial.

Lines 353-354: Is the “forearm” strategy correctly named? The authors state (and the figures show) that in this ‘group’, participants used both the forearm and the wrist. Referring only to the forearm as a ‘strategy’ may give a wrong impression. An idea would be to use a more vague term, such as ‘hybrid’ strategy. However, this is truly a minor comment, as the authors explicitly state that the forearm strategy involves both the wrist and the forearm.

Lines 434: Is there a reason that the factor ‘slope’ is not considered here?

6. PLOS authors have the option to publish the peer review history of their article (what does this mean?). If published, this will include your full peer review and any attached files.

Reviewer #1: No

Reviewer #2: No

Reviewer #3: No

---

## [Author Response · Author response to Decision Letter 0]

19 Nov 2021

Dear Editor,

First, let us thank you for your feedback as well as the reviewers’ comments that helped us to improve greatly our manuscript. Then, please, find in separate files: the two versions of our revised manuscript (the marked-up copy and its unmarked version), as well as our point-to-point response (in red color) to the reviewers’ comments. Note that together with the revised manuscript, we provide a new link (https://doi.org/10.5281/zenodo.5705206) to a directory containing the data, together with matlab processing codes and an explanatory “README FIRST MATLAB program.pdf” file.

We hope that this new version complies with the requirements for publication of our work in PLOS One.

Best regards,

Sylvain Famié, 

on behalf of my co-authors

---

## [Decision Letter · Decision Letter 1]

19 Dec 2021

PONE-D-21-08226R1Evidence for an internal model of friction when controlling kinetic energy at impact to slide an object along a surface toward a targetPLOS ONE

Dear Dr. Famié,

Thank you for submitting your manuscript to PLOS ONE. After careful consideration, we feel that it has merit but does not fully meet PLOS ONE’s publication criteria as it currently stands. Therefore, we invite you to submit a revised version of the manuscript that addresses the points raised during the review process. Specifically, one of the reviewers still has several comments and requests that the clarity of presentation should be improved. This also matches the impression of another reviewer, who also has problems to "fully follow the logic of the authors". I strongly recommend to work on theses issues and clarify the points raised by one of the reviewers.

We look forward to receiving your revised manuscript.

Kind regards,

Stefan Glasauer

Academic Editor

PLOS ONE

Reviewers' comments:

Reviewer's Responses to Questions

**Comments to the Author**

1. If the authors have adequately addressed your comments raised in a previous round of review and you feel that this manuscript is now acceptable for publication, you may indicate that here to bypass the “Comments to the Author” section, enter your conflict of interest statement in the “Confidential to Editor” section, and submit your "Accept" recommendation.

Reviewer #1: All comments have been addressed

Reviewer #2: All comments have been addressed

Reviewer #3: (No Response)

2. Is the manuscript technically sound, and do the data support the conclusions?

Reviewer #1: Partly

Reviewer #2: Yes

Reviewer #3: Partly

3. Has the statistical analysis been performed appropriately and rigorously? 

Reviewer #1: Yes

Reviewer #2: Yes

Reviewer #3: I Don't Know

4. Have the authors made all data underlying the findings in their manuscript fully available?

Reviewer #1: Yes

Reviewer #2: Yes

Reviewer #3: Yes

5. Is the manuscript presented in an intelligible fashion and written in standard English?

Reviewer #1: (No Response)

Reviewer #2: Yes

Reviewer #3: Yes

6. Review Comments to the Author

Reviewer #1: Thank you for addressing my comments.

I don't fully follow the logic of the authors, but based on thorough data analyses performed I want to believe that you have well thought this through and I respect your opinion. Probably the same question could have been addressed in more straight forward experiment requiring less convoluted analyses. I am not sure who will benefit from this work and unnatural experimental task, but I hope someone will.

The most relevant studies in the context of meaningful and practical motor tasks, like dexterous object manipulation, are still not even mentioned. Such studies have plenty of evidence for internal models of friction and clever strategies how they are used.

Reviewer #2: The authors have addressed all recommendations to my satisfaction. They closed gaps in their presentation by the inclusion of additional results and extension of the discussion section.

Reviewer #3: I thank the authors for their responses to the previous comments. I still believe that the results are of interest, however I am quite lost in the messages that the authors aim at conveying. The additional revisions have improved some parts but made some others either bulky or technical, while information about the analysis procedures are obscured or rather unclear. Below I would like to start first with some clarifications from my side on my previous comments. Then I provide my comments on the revised manuscript, as well as suggestions for improvements on the presentation and interpretation of the work.

Clarifications of some previous comments

The authors in their response letter state that they do not understand why I consider performance ‘stably suboptimal’ in some conditions. In particular, Figure 3, upper left panel, shows spatial error in the -10 deg condition. Here we see that the presented averages for each trial and material do not follow any consistent pattern. To be more precise, hoping that this will make clearer what I mean, for the Aluminium surface, the cube overshoots the target by ~8%, then a ‘correction’ occurs (trial 2) followed by a sequence of gradual undershoots (trials 3-5). Then participants overshoot again, before minimizing the error in the last 3 trials.

When it comes to the wooden surface, again in the same panel, participants systematically undershoot, sometimes more, sometimes less, until the last 3-4 trials, when their spatial error is around zero.

These results, particularly those for the Aluminium surface, show that performance is stably changing across trials, but that this change is not systematically in one direction (as, for instance, in the 0-deg condition). This does not necessarily mean that people do not learn, of course. The lack of an exponential fit may reflect an exploratory strategy, during which subjects try out different ways of hitting the object until they find the one that works ‘best’.

Alternatively, the use of the spatial error by the authors may be incomplete in capturing performance. Right now, the authors consider the error as the distance between the front part of the cube and the ‘target’s distance’ (page 11, lines 222-225). Considering that the triangular sticker and the respective marker are also not clearly evident in Figure 1, as the authors refer to that figure, I cannot tell whether it is possible that participants could exploit a range of possible endpoints that can be considered an accurate end-position. If this is the case, then the authors could consider highlighting on the graph which % range belongs to any part of the target area matching the front end of the cube.

In my previous review, I mentioned that performance for the wooden condition in the +10 slope does not improve over trials. At least seeing from the graph, I insist on that and cannot understand the authors’ response. As the authors also state, spatial error is similar between trials 1 and 10, and I guess between all other trials in that condition (wooden, +10). So, I guess they actually agree with my comment, namely that performance did *not* change across trials.

Comments

As suggested before, the authors now provide more p-values, but not in all instances. There are several places where they report that the p-value is below the alpha level (e.g., page 14, line 277). Although it is important to remind the reader about the alpha level, the p-value is not given. The p-values need to be provided explicitly (and the alpha-level can be provided as a reminder whenever necessary).

Following this, I would like to emphasize that the authors chose to sometimes correct for 12 comparisons, setting the alpha-level to 0.004. This is good when it comes to type 1 errors but it also shows that the design chosen by the authors together with the Bonferroni correction is conservative. Considering these, authors need to be more transparent about their procedures and results. Namely, a detailed overview of the statistical procedures should be provided in the Methods (not in Results), explaining *when* do they correct and for how many multiple comparisons. Right now, there are some explanations spread across the Results that makes it difficult to appreciate the procedure, focus on the findings, and appreciate the interpretations. In addition, reports of whether the data are normally distributed or not should be added, so that the applied statistical tests can be justified. And ideally, individual data points should be used for visualization, according to modern approaches about data transparency, which should assist the reader in understand the nature of the reported (no-)effects (see also a later comment).

I also have a question about the sample size. Was that determined on the basis of some power analysis or other procedure? If not, why is the sample size not even so that each of the [0, +10, -10 deg] and the [0, -10, +10 deg] groups have the same number of participants?

Presentation of testing

This remains quite confusing to me. I had to read some parts several times to understand what comparisons the authors make, and for what research questions.

For instance, in page 14, line 285, the authors refer to Figures 3 and 4 to explain the number of tests applied, but this implicit information is hard to digest. Further, in page 15, lines 315-317, there are tests between trial 1 of condition n and trial 10 of condition n-1, but there is no information about the difference of each of these trials from zero (a no-difference between these two trials does not mean that one or both of these are also no-different from zero).

A possible way to improve these, and other several instances across the manuscript, is to work on the structure of the Methods and Results. As an example, the authors could explain in the Methods what tests are they planning to report and with what (parametric or non-parametric) statistical procedures, using separate paragraphs, one for each testing procedure.

For instance, they could state something like: “We first wanted to test whether there were systematic spatial errors in the first trial of each condition. For this, we calculated spatial error in trial 1 of each participant and for each of the 12 conditions, and submitted these in a 3x2x2 ANOVA. Interactions were explored with further ANOVAs, and main effects with Bonferroni-corrected t-tests.”

Then, in another paragraph they could go on with something like: “We then wanted to test whether the spatial errors differed between the first and last trials of each block. Here, we examined with 12 paired two-sample t-tests, one per condition, whether the spatial error in trial 1 was different from that in trial 10….”

And then, in another paragraph: “In order to examine XXXX, we also compared spatial errors between the first trial of a given condition with the errors in the last trial of the previously presented condition. For this purpose, we conducted this and that test…”.

Of course, these are just suggestions. They may not capture completely what the authors intent to do but serve as ideas and the authors may find better ways in streamlining and explaining the procedure and testing they applied.

Results

There is a good deal of the manuscript devoted to the results shown in Figure 3. Though this makes sense, the authors subsequently show that there are differences in performance between the [0, +10, -10 deg] and [0, -10, +10 deg] groups. In page 15, line 312, the authors remind the reader about the two groups. Yet, they continue with reporting the average performance. This is confusing because the first sentence of the paragraph makes the reader think that the following results (say, of the -10 deg condition) refer to when that condition was presented right after the 0-deg condition (which, apparently is not true, but this becomes clear only later). Please make more explicit in the Results that you first report average performance for each condition; a sub-section in the beginning of the Results may help in clarifying this. Also, please explain why Figure 3 does not show data separately for each (25 and 50 cm) distance.

Page 16, lines 325-328: Shouldn’t this “quite successful” reparameterization be reflected in the comparison between trial 1 of that condition and trial 10 of the previously presented condition?

It is also not clear why the absence of motor adaptation speaks in favor of a reparameterization. Perhaps the authors need to be more explicit in what exactly they mean with ‘reparameterization’. But that there is no motor adaptation, at least the way that the authors compute it, does not necessarily mean that participants did the task successfully…the shown error of ~8% is something like 2 and 4 cm for the 25 and 50 cm conditions, which in my understanding this should be visually evident to the participants (unless the authors can show that the Weber’s fraction of such error perception is similar or larger than the errors reported here).

Such errors in trial 1 may also *not* be below the alpha level, and thus performance is considered ‘quite successful’ by the authors, either because the alpha level is set too low (due to the several comparisons planned by the authors) or due to variability across subjects (which is not shown and should be), or due to inappropriate testing (distribution of data).

Telling from the figures, one may even conclude that participants optimize energetic than accuracy factors. For instance, since the speed seems rather constant (most curves in Figure 3), it might be that people prefer to apply the same energy with their movement which leads to different initial cube speeds (depending on the material) and thus on different spatial errors. Obviously, this is not always the case, as nicely shown for the 0-deg condition, but can be an additional explanation of the rather variable spatial errors in several cases.

Page 16, lines 342-343: Please remind the reader about the purpose of this analysis. Why is the slope removed from the factors? What is the additional information compared to the previous analysis?

Page 18, line 387: What are the 24 tests that the authors imply here? And what are the four tests implied in line 390 of the same page? It is quite hard to follow.

Page 21, line 443: “appropriate movement re-parameterization”. I still do not get this. Could you please be explicit and show which results and tests you refer to? Do you compare trials 1 and 10, or trial 1 against zero (or something else)?

Minor

Please state explicitly the abbreviation of the International Society for Biomechanics (line 122, page 6).

Page 9, line 178: How is the cube’s movement onset defined?

Page 9, equation 1: What does the ‘min’ as a nominator represent? The minimum of what? This formula is confusing. Also, why do the authors use 40% as their threshold when using the MSI for deceleration?

Page 12, equation 5: a parenthesis is redundant after θwrist?

Page 19, lines 405-406: Do the authors mean that the speed needs to increase that much to achieve perfect performance? Please clarify.

Page 19, lines 407-408: I am sorry but I cannot follow this argument. Could you please clarify?

Page 29, line 643 (and elsewhere): Expressions such as “pretty good” are vague. How is “pretty good” defined? Please use objective language.

Yet, the authors state that spatial error was not different from zero, both in trial 1 and in trial 10. For this (and for several other comparisons), they do not report p-values in the text, but rather that the p-value is smaller than the set alpha level. Since the authors correct for multiple comparisons when looking at the spatial error in trial 1 of each of the 12 conditions, the set alpha is lower than the p-value. First, please report the p-values always, and remind the reader of the alpha level in the respective test.

7. PLOS authors have the option to publish the peer review history of their article (what does this mean?). If published, this will include your full peer review and any attached files.

Reviewer #1: No

Reviewer #2: No

Reviewer #3: No

---

## [Author Response · Author response to Decision Letter 1]

3 Feb 2022

The response to reviewers has been uploaded as a separate file.

---

## [Decision Letter · Decision Letter 2]

10 Feb 2022

Evidence for an internal model of friction when controlling kinetic energy at impact to slide an object along a surface toward a target

PONE-D-21-08226R2

Dear Dr. Famié,

We’re pleased to inform you that your manuscript has been judged scientifically suitable for publication and will be formally accepted for publication once it meets all outstanding technical requirements.

Kind regards,

Stefan Glasauer

Academic Editor

PLOS ONE

Additional Editor Comments (optional):

Reviewers' comments:

Reviewer's Responses to Questions

**Comments to the Author**

1. If the authors have adequately addressed your comments raised in a previous round of review and you feel that this manuscript is now acceptable for publication, you may indicate that here to bypass the “Comments to the Author” section, enter your conflict of interest statement in the “Confidential to Editor” section, and submit your "Accept" recommendation.

Reviewer #3: All comments have been addressed

2. Is the manuscript technically sound, and do the data support the conclusions?

Reviewer #3: Yes

3. Has the statistical analysis been performed appropriately and rigorously? 

Reviewer #3: Yes

4. Have the authors made all data underlying the findings in their manuscript fully available?

Reviewer #3: Yes

5. Is the manuscript presented in an intelligible fashion and written in standard English?

Reviewer #3: Yes

6. Review Comments to the Author

Reviewer #3: Thank you for your responses. There is just one typo that I spotted, line 334 "material"  "materials".

I am looking forward to your future work and best wishes.

7. PLOS authors have the option to publish the peer review history of their article (what does this mean?). If published, this will include your full peer review and any attached files.

Reviewer #3: No

---

## [Editor Report · Acceptance letter]

14 Feb 2022

PONE-D-21-08226R2 

Evidence for an internal model of friction when controlling kinetic energy at impact to slide an object along a surface toward a target 

Dear Dr. Famié:

I'm pleased to inform you that your manuscript has been deemed suitable for publication in PLOS ONE. Congratulations! Your manuscript is now with our production department. 

Kind regards, 

on behalf of

Prof. Dr. Stefan Glasauer 

Academic Editor

PLOS ONE